# Transcytosis via the late endocytic pathway as a cell morphogenetic mechanism

Renjith Mathew[1,†,¶], L Daniel Rios-Barrera[1,‡,¶] (iD), Pedro Machado[2,§], Yannick Schwab[2] (iD) & Maria Leptin[1,3,*] (iD)

## Abstract

**Plasma membranes fulfil many physiological functions. In polarized cells, different membrane compartments take on specialized roles, each being allocated correct amounts of membrane. The _Drosophila_ tracheal system, an established tubulogenesis model, contains branched terminal cells with subcellular tubes formed by apical plasma membrane invagination. We show that apical endocytosis and late endosome-mediated trafficking are required for membrane allocation to the apical and basal membrane domains. Basal plasma membrane growth stops if endocytosis is blocked, whereas the apical membrane grows excessively. Plasma membrane is initially delivered apically and then continuously endocytosed, together with apical and basal cargo. We describe an organelle carrying markers of late endosomes and multivesicular bodies (MVBs) that is abolished by inhibiting endocytosis and which we suggest acts as transit station for membrane destined to be redistributed both apically and basally. This is based on the observation that disrupting MVB formation prevents growth of both compartments.**

**Keywords** cell polarity; _Drosophila_; endocytosis; membrane traffic; sorting
**Subject Categories** Development; Membranes & Trafficking
**The EMBO Journal (2020) 39: e105332**

## Introduction

Most cells have specialized plasma membrane domains that serve dedicated physiological purposes. For instance, epithelial cells have an apical and a basal domain separated by adherens junctions and facing different parts of the body. Membrane and proteins are allocated to these domains in a way that is commensurate with their functions. For example, absorptive epithelia have massively enlarged apical domains organized in microvilli, and photoreceptor cells form specialized membranous outer segments for the light-sensing rhodopsins. Errors in the proportions of membrane domains can have harmful consequences for organ function (Wodarz _et al_, 1995; Wilson, 2011; Zang _et al_, 2015). Therefore, the mechanisms that balance plasma membrane distribution are crucial for morphogenesis and tissue homeostasis.

Lipids are synthesized in the ER and trafficked to the plasma membrane via the Golgi apparatus or directly through ER–plasma membrane contact sites (Holthuis & Menon, 2014). Membrane delivery depends on fusion machinery, including SNAREs, and tethers such as the exocyst complex (Wu & Guo, 2015; Saheki & De Camilli, 2017). Due to technical limitations of direct labelling of membrane lipids _in vivo_, most studies addressing membrane trafficking follow cargo proteins. These are sorted generally at the trans-Golgi network using receptors like components of the adaptor protein 1 (AP-1) complex (Guo _et al_, 2014). Rab proteins can also participate in directing polarized secretion (Lerner _et al_, 2013; Bellec _et al_, 2018).

Material can also be passed from one domain to the other by transcytosis, which can occur either from apical to basal or vice versa, e.g., IgG and IgA in the gut (Fung _et al_, 2018). The main role described for transcytosis is to transport cargo from one side of an epithelium to the other. However, redistribution of plasma membrane may also be used for other purposes, including cell morphogenesis (Pelissier _et al_, 2003; Soulavie _et al_, 2018). The trafficking routes and the delivery mechanisms are not understood for these processes, nor is it known whether they are isolated special cases.

A cell type with sophisticated morphogenesis and pronounced specialization of membrane domains is the tracheal terminal cell in insects, which transports oxygen through a branched network of subcellular tubes. Tracheal terminal cells form long hollow branches, with the apical compartment forming the luminal membrane of each branch and the basal compartment facing the body's inner cavity. This architecture is formed by mechanisms

---

1 Directors' Research Unit, European Molecular Biology Laboratory, Heidelberg, Germany
2 Electron Microscopy Core Facility, European Molecular Biology Laboratory, Heidelberg, Germany
3 Institute of Genetics, University of Cologne, Cologne, Germany
  *Corresponding author. Tel: +49 6221 8891-5101; E-mail: mleptin@uni-koeln.de
  ¶These authors contributed equally to this work and are listed alphabetically
  †Present address: National Institute for Science, Education and Research, Bhubaneshwar, India
  ‡Present address: Institute for Biomedical Research, Universidad Nacional Autónoma de México, Mexico City, Mexico
  §Present address: Centre for Ultrastructural Imaging, King's College London, London, UK
  [Correction added on 20 July 2020, after first online publication: Projekt Deal funding statement has been added.]

shared by other lumen-forming tissues like endothelial cells and Madin–Darby canine kidney (MDCK) cells grown in 3D (Sigurbjörnsdóttir *et al*, 2014). These mechanisms have been widely studied in *Drosophila* larval tracheal cells (Ghabrial *et al*, 2011; Baer *et al*, 2012; Schottenfeld-Roames & Ghabrial, 2013; Jones *et al*, 2014; Rios-Barrera *et al*, 2017), where they are, however, usually limited to endpoint phenotypes with only short-term live imaging possibilities.

By contrast, cell morphogenesis and tube formation can easily be observed live in the embryo (Gervais & Casanova, 2010; JayaNandanan *et al*, 2014; Okenve-Ramos & Llimargas, 2014; Ricolo *et al*, 2016), where the first branch forms. The subcellular tube develops by an invagination of the apical compartment, and membrane is added throughout the length of the invaginating tracheal tube (Gervais & Casanova, 2010). The tube grows in unison with the basal domain of the cell, and as the branch and its tube extend, trafficking markers are seen throughout the cell, often associated with the tube (Gervais & Casanova, 2010; JayaNandanan *et al*, 2014; Schottenfeld-Roames *et al*, 2014). Studies in larvae show that failures in endocytosis, exocytosis and secretion can all affect the growth of both membrane domains as judged by reduced cell branching (Jones *et al*, 2014; Schottenfeld-Roames *et al*, 2014; preprint: Rios-Barrera *et al*, 2017). However, only the loss of endocytic components has been associated with defects in subcellular tube size and architecture (Schottenfeld-Roames *et al*, 2014). How these vignettes of knowledge fit into a picture of an integrated membrane delivery mechanism that balances apical and basal delivery is not known. Here, we address this by perturbing membrane dynamics and looking at the redistribution of membrane and protein markers during terminal cell development in the embryo.

# Results

## Organization of membrane domains during subcellular tube morphogenesis

Tracheal terminal cells elongate with their subcellular, apical tube and outer, basal membrane compartment growing at the same rate (Fig 1A–D). Since lipids are produced in the ER, and membrane proteins and secreted components of the tube must pass through the Golgi, these organelles are likely critical for the expansion of the plasma membrane. To study their distribution, we used KDEL::RFP as a maker for ER and the Golgi-targeting sequence of β-1,4-galactosyltransferase fused to GFP (GalT::GFP) as a trans-Golgi marker, together with CD4::mIFP, a membrane reporter (Yu *et al*, 2015) that is enriched in the plasma membrane but also seen in other subcellular membrane compartments. Both organelles are present in the cell body, throughout the length of the growing branch and in the growth cone ahead of the tube (Fig 1C and E).

We also noticed accumulations of CD4::mIFP-labelled membrane material in the space between the extending filopodia of the growth cone and the tip of the growing tube (Fig 1C). A similar accumulation in this position has been reported for Par3 (which is mainly associated with the apical membrane) but suspected to be an artefact of Par3 overexpression (Gervais & Casanova, 2010). We nevertheless considered the possibility that these structures might be part of the extending tube, or perhaps nascent plasma membrane, and analysed them with a range of membrane markers. They were also seen with other general membrane reporters (Appendix Fig S1A–C), but not with markers considered to be selective for the plasma membrane, such as the Pleckstrin homology domain (PH) of PLCδ fused to GFP or mCherry. These labelled only the tube and the basal membrane of the cell (Fig 1F; Appendix Fig S1B and C; Movie EV1). This suggests that the material does not correspond to plasma membrane.

The presence of Par3 in this region is therefore intriguing, and we tested other characteristic apical membrane-associated proteins for their distribution. A GFP inserted into the *crumbs* (*crb*) locus, Crb::GFP, was seen in its normal location at the tube membrane and also with the CD4 vesicles near the tip of the cell (Fig 1G), similar to an overexpressed construct (Appendix Fig S1D). Par3 and Par6 also localized to CD4 vesicles ahead of the tube (Fig 1H). On occasions where different polarity markers were associated with the same vesicle, their overlaps remained partial (Fig 1I). The fact that Crb::GFP fluoresces indicates that it is not recently synthesized [GFP maturation time is more than 1 h (Balleza *et al*, 2018)] and that this compartment therefore does not represent an intermediate along the biosynthetic pathway from the Golgi to the plasma membrane. Instead, we conclude that most likely these structures are endosomes that arise from the apical membrane. This conclusion was supported by high time resolution imaging of Par3 and CD4. The Par3 ahead of the tube appeared to originate from the tube and move to the tip (Appendix Fig S1E; Movie EV2), as previously reported (Gervais & Casanova, 2010).

To understand the nature of this domain, we used serial section electron tomography. To generate an atlas of organelle distribution throughout the length of the cell, we initially screened serial sections manually to identify terminal cells. Though feasible, this labour-intensive workflow allowed only limited analyses. To avoid screening using EM, we turned to correlative light and electron microscopy (CLEM), building on previous protocols that preserve the signal from fluorescent proteins (Nixon *et al*, 2009; Kukulski *et al*, 2011). Embryos expressing KDEL::RFP and Par3::YFP under *btl-gal4* were fixed and serially sectioned to cover at least one full embryonic segment (200 sections of 300 nm). The fluorescent signal allowed rapid identification of the terminal cells to be imaged by high-resolution electron tomography (Fig EV1).

We first obtained low-resolution overviews of cell morphology (Fig 2A and C) and then used higher-resolution tomograms to manually trace organelles (Fig 2B and D). The tomograms confirmed the distribution of ER and Golgi we had observed by light microscopy, with both organelles spread throughout the length of the cell (Fig 2A–A″). We also found a range of vesicles with distinctive electron densities, sizes and distributions (Fig 3A–C). Many MVBs and large electron-dense vesicles were seen close to the tip of the tube (Fig 3A and B; Movie EV3). This, together with the presence of Crb, Par3 and Par6 in this location, suggests extensive membrane trafficking or active recycling events throughout the cell and particularly in the growing tip.

## The role of endocytosis in terminal cell morphogenesis

The basal plasma membrane and the subcellular tube grow in unison. Hence, there must be a mechanism to balance the delivery of membrane between the two domains. There are in principle two

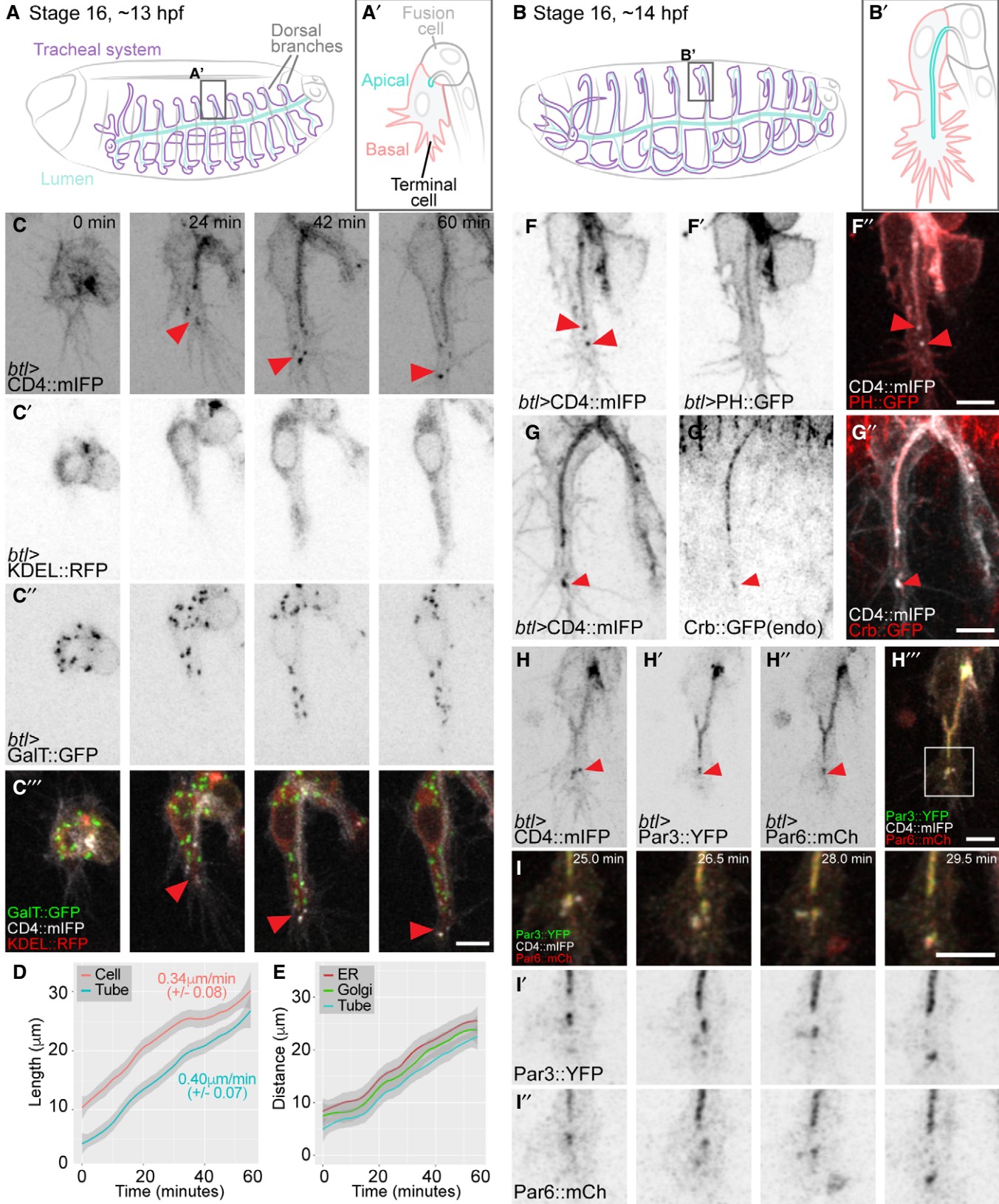

**Figure 1.**

◄

**Figure 1. Membrane organization during tube morphogenesis.**

A, B Diagram of the organization of the tracheal system at the onset of terminal cell branching (A) and 1 h later (B).

C–C‴ Terminal cell expressing reporters for membrane (C), ER (C′) and Golgi apparatus (C″).

D Mean and 95% confidence interval of tube and branch length during the first hour of growth. Numbers represent the rate of growth estimated from the slope of the curve ± SD, P = 0.39, Mann–Whitney U-test, n = 6.

E Distance of the tip of the tube, and the leading ER and Golgi accumulations from the cell junction that defines the base of the cell. Line and shading represent mean and 95% confidence interval, n = 4.

F–I Terminal cells expressing the general membrane marker CD4::mIFP in combination with different membrane or polarity markers: PH::GFP, a PIP$_2$ sensor commonly used as apical plasma membrane marker (F′), an endogenously GFP-tagged Crb (G′), polarity proteins Par3::YFP (H′) and Par6::mCherry (H″). The boxed region in (H‴) is shown at higher magnification in (I) over 4 time points.

Data information: Red arrowheads: CD4::mIFP vesicles and associated markers at the tip of the cell. Anterior is left, and dorsal is up. Scale bars: 5 μm.

Source data are available online for this figure.

ways to achieve this: (i) membrane is delivered from the ER and Golgi directly and in the correct measure to each compartment. (ii) ER and Golgi deliver membrane to one compartment, and part of this material is then retrieved and transported towards the other via transcytosis. These models can be distinguished experimentally, because the latter requires endocytosis from the plasma membrane for the shuttling. To test this, we blocked endocytosis using a temperature-sensitive allele of *dynamin*, *shibire*[ts] (Koenig & Ikeda, 1989), which can be inactivated within 15 min by shifting the embryos to 34°C.

We blocked dynamin at the onset of tube formation in cells expressing PH::GFP, a construct commonly used as a marker for apical membrane but which is also visible in the basal plasma membrane (Fig 4A and B). Unlike control cells, where basal and apical membrane domains expanded at similar rates (Fig 4A, Movie EV4), cells in which dynamin was inactivated failed to grow properly. *shibire*[ts] cells showed an excessive increase in membrane material inside the cell whereas the basal membrane failed to grow (Fig 4B′), leading to a shift in the proportions of membrane on each domain. In control cells, the proportion of fluorescent material in each compartment remains constant during cell growth (12% in the apical versus 88% in the basal domain, ± 2 SD), whereas it gradually increased in *shibire*[ts] cells, reaching up to 35% in the apical and 65% in the basal ± 10 SD (Fig 4C). Blocking dynamin function in older cells where the basal membrane and the tube had already extended led to the accumulation of the marker throughout the length of the tube (Fig 4E, Movie EV4). The defects in cell and tube growth were reversible: shifting the embryos back to the permissive temperature restored the expansion of the basal membrane and resulted in partial or complete resolution of the membrane accumulation in the tube domain (Fig 4B″, Movie EV4).

While it was evident that the outer membrane domain expansion was strongly reduced after blocking endocytosis, it was less clear whether the internal membrane pool corresponded to the normal amount that had simply been compacted within a smaller volume, or whether more apical membrane was present. The finding that total fluorescence intensity (basal plus apical) of the cell increased at the normal rate after dynamin inactivation (Fig 4D) shows that plasma membrane synthesis and delivery *per se* were not affected. Our measurements indicate that upon dynamin inactivation, a similar amount of membrane material as would normally have been added to the basal domain had instead accumulated in the subcellular tube, in addition to the material that is normally delivered there.

We confirmed the identity of the material within the cell as apical membrane by the presence of Crb and of Gasp::GFP, a protein secreted into the tracheal lumen. Crb colocalized with the membrane reporter within the cell (Appendix Fig S2A, B, D and E), and it remained associated with the tube after recovery (Appendix Fig S2C and F). Gasp::GFP formed complex ramifications that sprout from the subcellular tube (Fig 4F and G). These results show that dynamin, and therefore most likely endocytosis, is needed for the correct allocation of the appropriate amounts of membrane to the basal and apical compartments. Specifically, membrane is delivered to the apical domain, and it remains there in the absence of endocytosis, while insufficient or no membrane finds its way to the basal domain. According to this model, endocytosis should be more prominent at the apical domain. Consistent with this, both the clathrin light chain (CLC) and dynamin itself were present throughout the cell but they were more abundant around the tube than in the basal membrane (Fig 4H and I). This was seen in EM images, where we found a higher density of endocytic events at the apical than at the basal membrane (Fig 3D).

To test whether raised levels of Crb were responsible for the excessive apical membrane, as reported in other contexts (Wodarz *et al*, 1995; Pellikka *et al*, 2002; Schottenfeld-Roames *et al*, 2014), we knocked down *crb* (Fig EV2A and B). This approach reduced the level of Crb to ~50% of that seen in control cells (Fig EV2C–E). However, this reduction did not alleviate the accumulation of apical membrane. We conclude that blocked endocytosis directly interferes with bulk apical membrane retrieval.

We also considered other potential reasons for the lack of basal membrane growth. Because endocytosis is involved in receptor tyrosine kinase signalling (Villaseñor *et al*, 2016) and FGF signalling is required for terminal cell growth, the growth defect could be due to impaired FGF signalling. To test this, we quantified ERK phosphorylation, a signature of FGF receptor activation (Gabay *et al*, 1997). Blocking dynamin led to a slight increase in di-phospho-ERK. This was not simply due to the temperature increase, as it was not observed in control embryos shifted to 34°C (Fig EV2F–K). Thus, blocking dynamin function does not prevent FGF signalling activation, and if anything, it results in a slight increase in the activation of ERK.

Having excluded other explanations, we suggest that the lack of basal membrane growth and overgrowth of tube membrane are functionally connected and that the normal balance between the two requires endocytosis. The simplest scenario would be transport of apical membrane material to the basal domain by transcytosis.

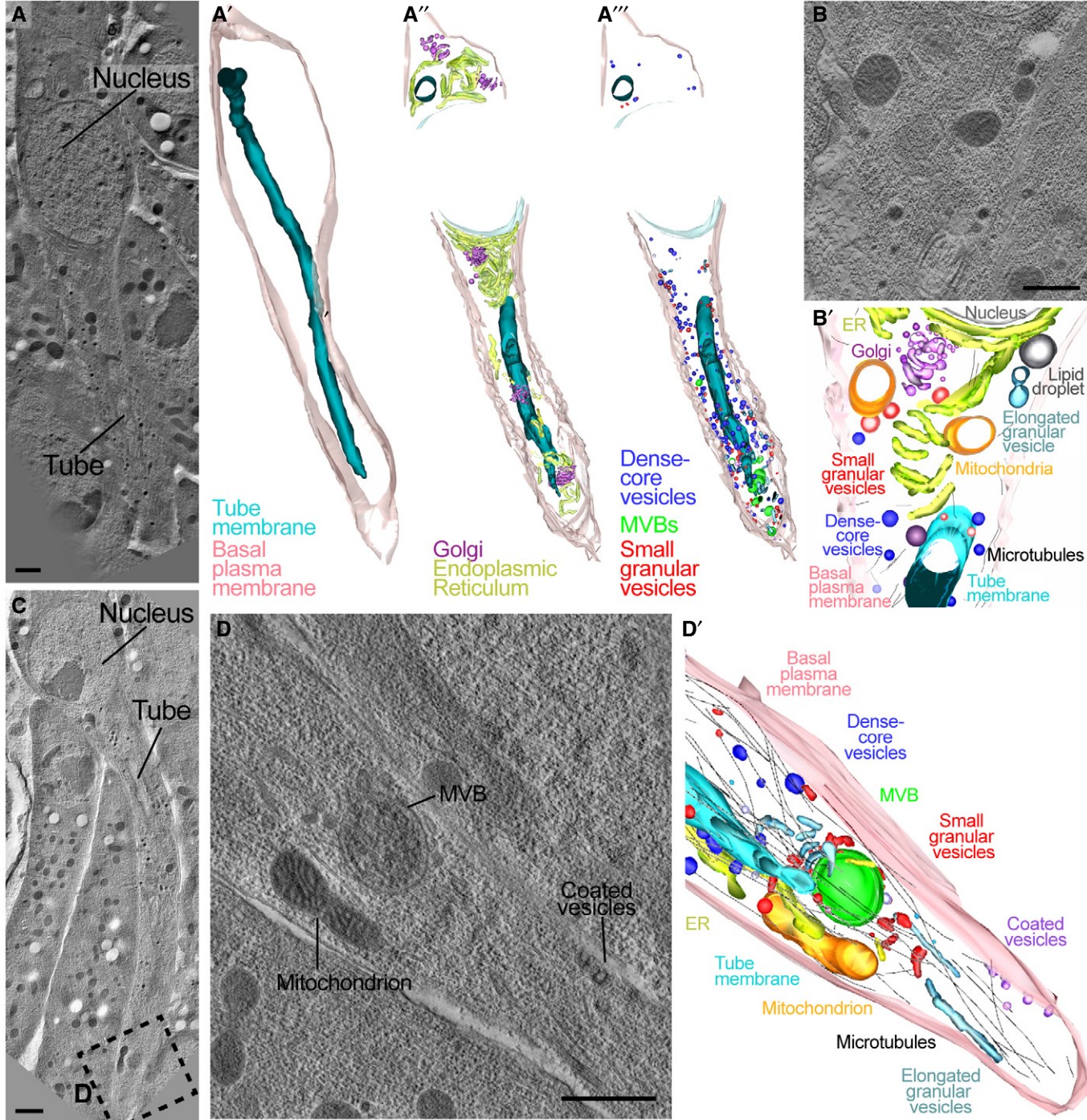

**Figure 2.  Reconstruction of organelles in a terminal tracheal cell by electron tomography.**

A–D   TEM tomograms from high-pressure frozen embryos.
A–A′   Low magnification (2.5 nm voxels) tomogram and 3D reconstruction from 10 serial 300-nm sections that cover one terminal cell.
A″–A‴   3D reconstructions derived from higher-resolution tomograms showing different organelles in the cell.
B–B′   Example of an individual high-resolution (0.78 nm voxels) tomograms used to generate 3D reconstructions.
C   Low-resolution tomogram of another cell; the dotted box is magnified in (D).
D   High-resolution tomogram.
D′   Model derived from 4 serial 3D reconstructions including the one shown in (D).

Data information: Scale bars: 1 µm (A, C); 500 nm (B, D).

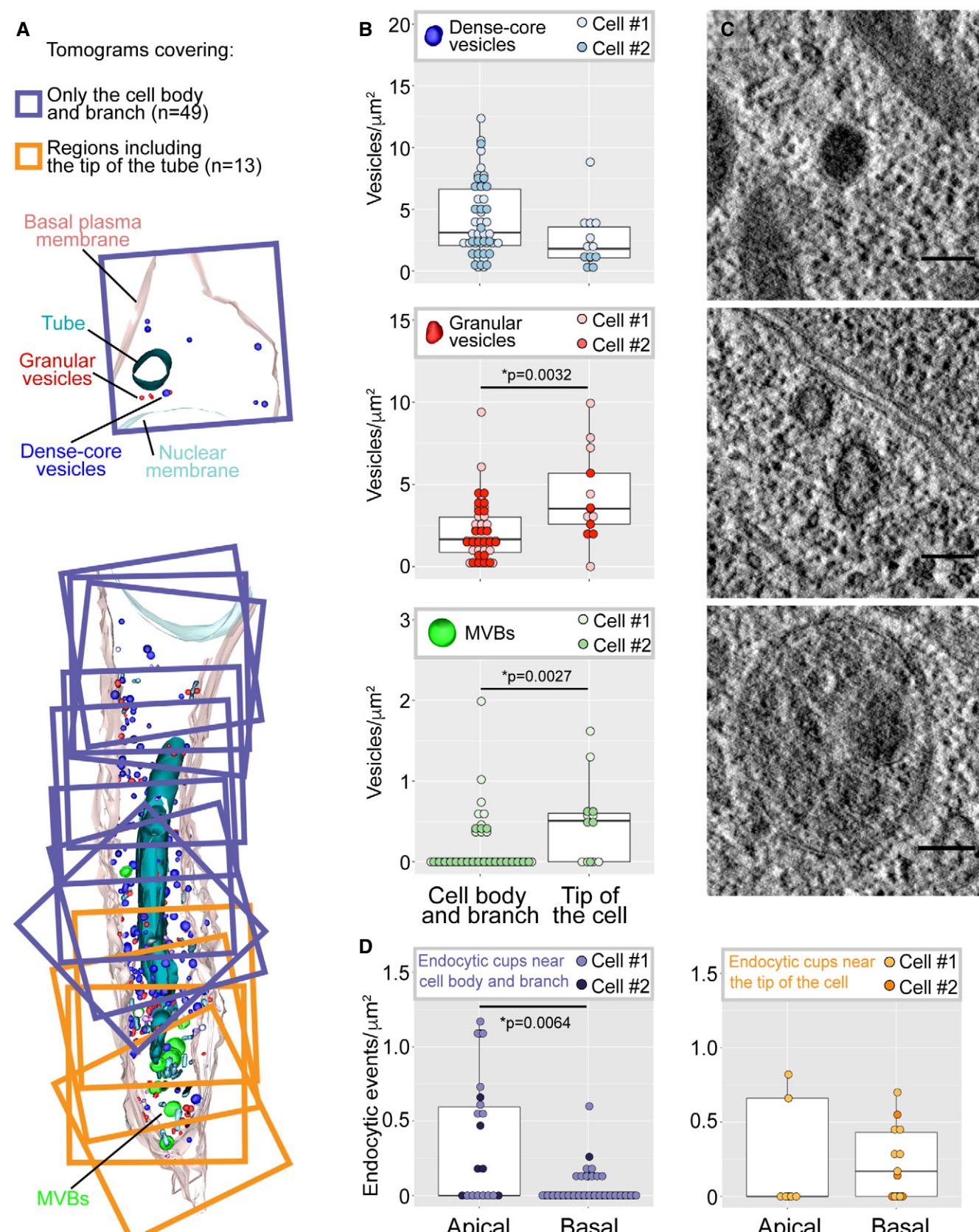

**Figure 3.**

**Figure 3.  Representative vesicle types and their distribution in the terminal cell.**

A   Cell model composed of superimposed 3D reconstructions of high-resolution TEM tomograms covering different regions of the cell. Each square is one 3.2 × 3.2 × 0.17 µm 3D reconstruction. The tomograms were assigned to two categories for quantitative evaluation: those including the tip of the tube (orange) and the rest (purple).

B   Distribution of three commonly seen classes of vesicles, expressed as their density per area of cytoplasm screened, in the two regions of the cell. Box plots represent median, interquartile range (IQR) and IQR*1.5 below and above the IQR. *n* = 62 tomograms from two different cells.

C   Representative examples for each vesicle class are shown next to the quantification plots.

D   Distribution of endocytic events at the apical or basal membrane compartment, expressed as density per membrane surface screened. Left shows tomograms corresponding to the cell body and branch, and right shows tomograms corresponding to the tip of the cell. Box plots represent median, interquartile range (IQR) and IQR*1.5 below and above the IQR. *n* = 48 tomograms from two different cells.

Data information: Significance was determined using Mann–Whitney U-test. Scale bars: 100 nm.
Source data are available online for this figure.

## Membrane morphology and distribution in control and *shibire^ts* cells

To see the effect of blocking dynamin on the membrane or other compartments within the cell, we used electron tomography. Control cells had smooth tube membranes, with apical extracellular matrix (aECM) in the lumen of older cells. This appeared as long fibres curling inside the tube and as electron-dense depositions adjacent to the plasma membrane (Fig 5A). In cells fixed after 15 min of dynamin inactivation, the tube membrane appeared largely similar to the control (Fig EV3A and B), consistent with the minor effects on cell morphology seen in live observations (Fig EV3C). However, we also found bulges protruding from it into the cytosol (Fig EV3B). These were also visible after 1 h of dynamin inactivation, where additionally the tube membrane appeared highly irregular (Figs 5B and EV3D and E, Movie EV5). We interpret these irregularities as endocytic events in which scission from the membrane had failed to occur, and indeed in several instances, these structures were surrounded by particulate electron-dense structures resembling clathrin coats (Fig 5B″).

After 2 h of dynamin inactivation, the morphology of the tube membrane was severely affected. The cells contained complex ramifications of the tube membrane together with its aECM (Fig 5C and F), which resembled the structures we had seen by light microscopy using Gasp::GFP (Fig 4G). The cells also showed more dramatic defects. Some invaginations from the tube membrane could be traced all the way to the basal plasma membrane, with which they were clearly contiguous (Figs 5D and EV3F, Movie EV5). All cells had instances of large sheets of membrane connecting different parts of the tube and the plasma membrane. These sheets did not represent auto-cellular junctions as seen in other tracheal cells (Francis & Ghabrial, 2015) since they contained neither Ecad nor FasIII (Appendix Fig S2G–J). In several places, the space between the two apposed plasma membrane sheets contained material resembling the content of MVBs (Figs 5B and E, and EV3E). If these sheets are the result of fusion events between invaginations emanating from the apical and basal plasma membrane, then these events must happen rarely and, when they do happen, rapidly, since we found no clear intermediates in our tomograms, such as apical invaginations reaching to the basal plasma membrane. However, we saw several instances of long invaginations from the apical membrane and occasional ones from the basal membrane, and in one cell, the surfaces of two such invaginations came within 500 nm of each other (Fig 5E, Movie EV5).

## Distribution of basal cargo in control and *shibire* cells

If the basal compartment is derived from the apical compartment by endocytosis followed by transcytosis, then general secretion from the ER/Golgi should be directed predominantly towards the apical compartment. This should also apply for transmembrane proteins that are destined for the basal membrane. If secretion follows the route we postulate, blocking dynamin should lead to basal cargo accumulating in the apical compartment before it can reach the basal domain. Hence, we looked at the transport of two known basal transmembrane proteins expressed in tracheal cells, the FGF receptor (FGFR) Breathless and the integrin β-chain Myospheroid (βPS-integrin).

In control cells, both proteins mostly localized to the basal filopodia, although FGFR was also seen in large puncta near the tip of the cell (Fig 6A and C). In *shibire^ts* embryos, both were abnormally accumulated at the apical membrane (Figs 6B and D–G, and EV2F″–J″ and L).

These findings support our conclusion that the initial delivery of membrane from the Golgi goes mainly to the apical domain and fails to be redistributed basally in the absence of endocytosis. An alternative interpretation for the apical mislocalization would be that the defect in *shibire^ts* cells was not one of the failed retrievals of basal proteins, but that cargo was misdirected immediately after the Golgi upon dynamin inactivation, as observed in a mammalian model (Deborde *et al*, 2008). If this is the explanation for the abnormal localization of βPS-integrin and FGFR, restoring dynamin function should not restore their proper localization. Conversely, if FGFR and βPS-integrin mislocalization was the result of failed membrane endocytosis at the tube, restoring endocytosis should correct their faulty localization. We observed by live imaging (Fig 6B, Movie EV6) and also by immunostainings (Fig 6F, Appendix Fig S2K–M) that after restoring dynamin function, FGFR distribution to the basal membrane was re-established (Fig 6B and F; Appendix Fig S2M; Movie EV6). These results support the model where basal proteins are delivered to the apical membrane first and are then transcytosed to the basal membrane by dynamin-mediated endocytosis.

## Vesicle trafficking and conversion

Our observations point to a role for apical-to-basal transcytosis in coupling the extension of apical and basal membrane domains. The ideal way of investigating this experimentally would be to follow the path of plasma membrane lipids by live imaging. Unfortunately,

no suitable probes exist that allow this in the developing tracheal system *in vivo*. Protein markers as proxies for membranes are not suitable to follow the full path because they are subject to sorting along the vesicle transport path. Instead, we studied individual segments of the potential path by imaging the movement of markers

for well-defined vesicular membrane compartments at high temporal resolution. Using CD4::mIFP, we found a large number of vesicles moving rapidly along the main axis of the growing terminal cell (Fig 7A, Movie EV7), often arising in the proximity of the tube and later moving towards the growing tip of the cell. Vesicle tracking

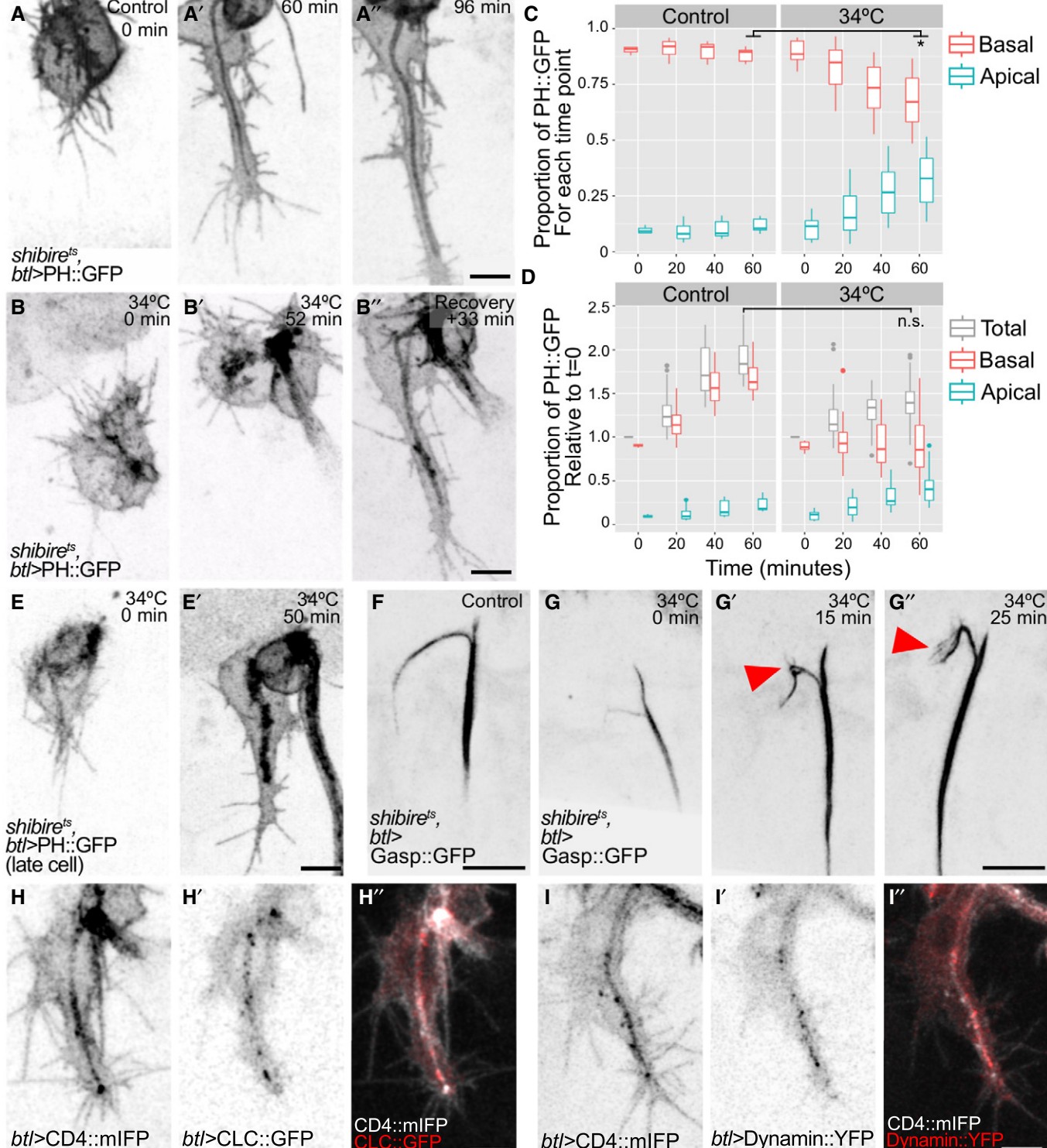

Figure 4.

**Figure 4. The role of endocytosis in terminal cell growth.**

A–E   Distribution of the plasma membrane reporter PH::GFP in control cells (A–A″) and in cells where dynamin activity had been blocked using a temperature-sensitive allele of *dynamin* (*shibire*[ts], B, E). In (B), it was inactivated at the onset of tube formation, whereas in (E), it was inactivated ~15 min after the tube had begun to form. (C–D) Distribution of PH::GFP fluorescence intensity in control and *shibire*[ts] cells. Data from 1- to 2-min interval time lapses were collected in windows of 20 min each (except for t = 0). Box plots represent median, interquartile range (IQR) and IQR*1.5 below and above the IQR. (C) Proportion of signal in the apical and in the basal membrane compartment over time in control cells (n = 4) and also in cells where dynamin was inactivated (n = 5), *P = 0.015, Mann–Whitney U-test. (D) Total fluorescence intensity of PH::GFP over time in controls (n = 4) and in cells where dynamin was inactivated (n = 5), n.s.: P = 0.063, Mann–Whitney U-test.

F, G   Distribution of the luminal reporter Gasp::GFP in control cells (F) and in cells where dynamin was inactivated (G). Arrowheads point to protrusions sprouting from the lumen of the terminal cell.

H, I   Distribution of the general plasma membrane marker CD4::mIFP in combination with the clathrin light chain fused to GFP (CLC::GFP, H) and with Dynamin::YFP (I).

Data information: Scale bars: 5 μm.
Source data are available online for this figure.

revealed a net distal displacement regardless of where the vesicles originated (Fig 7B and C). This was not a consequence of cell elongation taking place in the same direction since the vesicles moved faster than the rate of tube elongation (Fig 7D).

Vesicles first appearing in proximity to the tube suggested that they might be derived from the apical membrane, as was also indicated by the movement of the Par3-positive vesicles (Appendix Fig S1). We simultaneously imaged CD4::mIFP with GFP fused to the rat atrial natriuretic factor signal peptide (ANF::GFP), an apically secreted cargo (Tsarouhas *et al*, 2007). Many CD4 vesicles were positive for ANF::GFP, suggesting that they emerged from the tube (Fig EV4A and B). In conclusion, we see extensive distal vesicle movement, and at least some of the vesicular membrane appears to be derived from the subcellular tube. To test whether these structures constitute part of the endocytic pathway, we analysed Rab5 distribution as a marker for early endosomes and found it was visible as discrete puncta around the tube, often localizing to CD4 vesicles. The association between Rab5 and CD4 vesicles was transient, with Rab5 being gradually lost from the vesicles over a few minutes (Fig 7E). This suggested that Rab5 endosomes were progressing into later endocytic compartments.

Early endosomes can be recycled back to the plasma membrane and this pathway is often employed to transcytose material from one compartment to another, in a process that relies on Rab11 (Pelissier *et al*, 2003; Bryant *et al*, 2010; Soulavie *et al*, 2018). If this pathway was used in the process we describe here, we would expect to see Rab11 associated with the Rab5-positive structures near or at the CD4 vesicles. However, we found that both overexpressed Rab11 (this work and Gervais & Casanova, 2010) and endogenously tagged Rab11 formed clouds of small puncta that surrounded the tube and did not overlap with CD4 vesicles travelling to the tip (Fig EV4C and D).

Early endosomes can also undergo fusion and maturation into late endosomes. These steps involve replacement of Rab5 by Rab7 (Gillooly *et al*, 2001). Imaging Rab5, Rab7 and CD4 simultaneously showed several instances of CD4 vesicles containing both Rab5 and Rab7, and eventually losing Rab5 while retaining Rab7 (Fig EV4E and F). Rab5-to-7 conversion results in the recruitment of ESCRT components, leading to MVB formation and afterwards to the transition into lysosomes. We imaged a range of markers for different steps of the pathway: the FYVE domain of the ESCRT-0 component Hrs (FYVE::GFP) for early endosomes and MVBs; the lysosomal permease Spinster (Spin::RFP), recruited in MVBs; and dLamp1 for lysosomes (Johnson *et al*, 2015; Riedel *et al*, 2016). Similar to Rab5, FYVE::GFP formed small puncta throughout the length of the tube

(Fig 7F), but it was more prominently associated with the large CD4-positive structures at the tip of the cell. Spin::RFP, dLamp::mCherry and Rab7 were seen almost exclusively in association with CD4 vesicles at the tip of the cell (Figs 7F and H, and EV4G). Given that the early markers Rab5 and FYVE::GFP were seen throughout the length of the cell, whereas the late markers Rab7, Spin and dLamp were restricted to the tip, we conclude that tube-derived endosomes enter the late endosomal pathway during their movement towards the tip of the cell. This is consistent with our EM analyses where we found a polarized MVB distribution towards the tip of the cell.

To understand the relationship between the tube-derived vesicles, the endocytic pathway and the apical membrane determinants at the tip of the cell, we studied the distribution of apical polarity proteins relative to late endocytic markers. FYVE::GFP, Rab7 and dLamp1 were often seen in proximity to apical proteins (Figs 7G and EV4G and H), suggesting that endocytic vesicles carrying cargo proteins are converted from early to late endosomes as they are displaced towards the tip of the cell.

### Dependence of large intracellular membrane structures on endocytosis

The results so far suggest that the late endosomes at the tip of the cell are sustained by material endocytosed at least in part from the tube membrane. If that was the case, blocking endocytosis should affect them. Quantification of FYVE::GFP-positive vesicles showed that while control cells contained around 1–4 large FYVE::GFP-CD4 vesicles at any given time, cells where dynamin was inactivated had very few, often none, of these vesicles (Fig 8A–E). Upon recovery, the number of FYVE::GFP-CD4 vesicles was re-established (Fig 8F–H) and even increased beyond the control conditions. We obtained similar results with Rab7::YFP and Spin::RFP (Fig EV5A–F). This indicates that the large membrane accumulations at the tip of the growing tracheal cell are formed from material delivered by endocytosis. This was consistent with our experiments on Crb and FGFR distribution: both were seen in vesicles in control cells, and these vesicles disappeared upon dynamin inactivation but reappeared during recovery (Fig 6B).

### Dependence of cell growth on MVB

If the destination of the majority of material endocytosed from the tube membrane is the compartment of late endosomal vesicles at

the tip of the cell, the question arises where the membrane material moves from these structures, and whether any of it is delivered to the basal side. MVBs have been shown as a route for delivery of membrane and cargo to the plasma membrane (Zhang & Schekman, 2013; Dong *et al*, 2014a). Since in our experiments MVBs seem to

act as collection points for apical and basal cargo, they may act as a hub for membrane redistribution to the basal membrane domain. To test this, we interfered with the proper biogenesis of the MVBs. MVB formation requires the function of Shrub (Shrb), the *Drosophila* homolog of ESCRT-III component Snf7, and Shrb::GFP

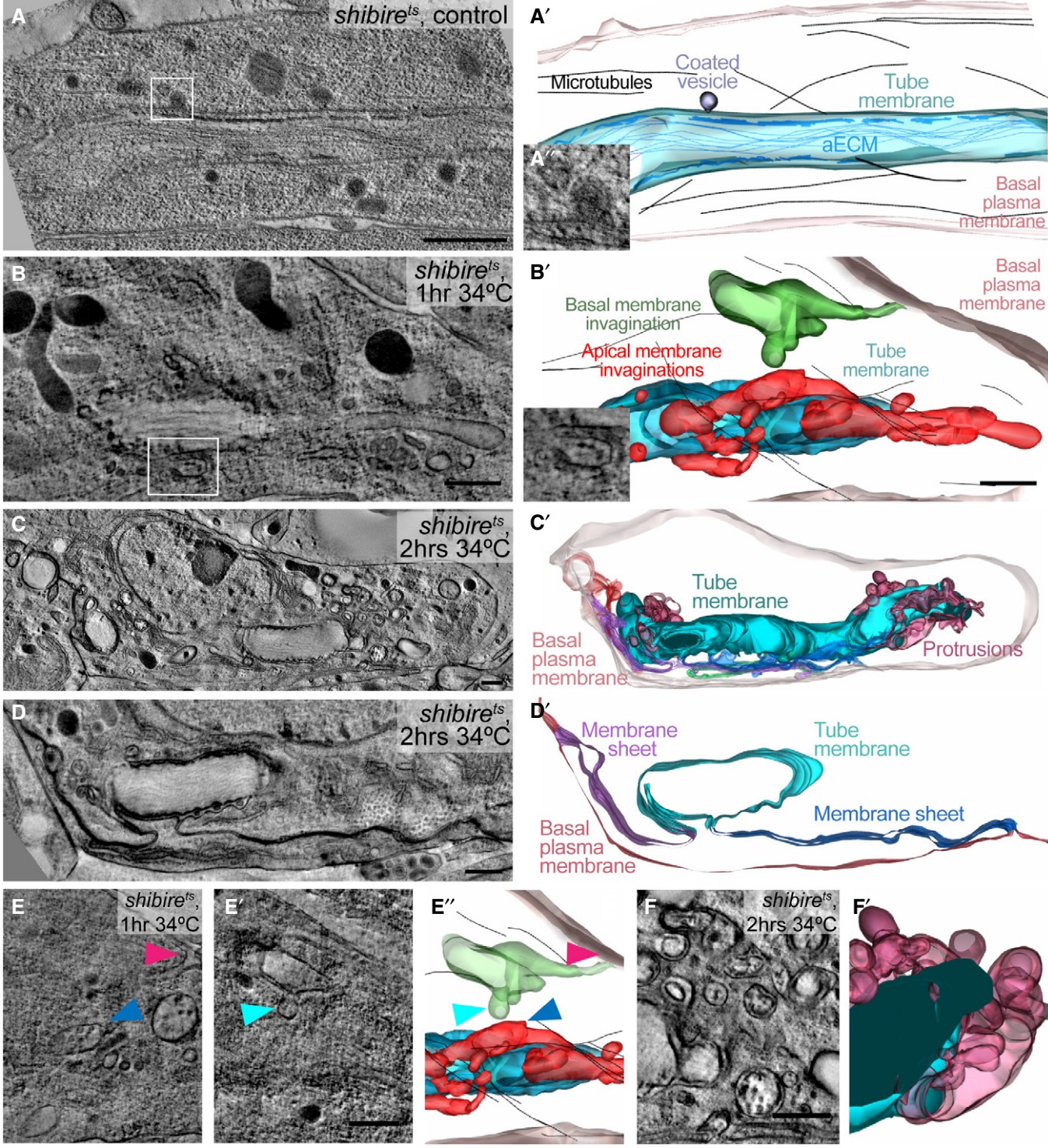

**Figure 5.**

**Figure 5.   Effects of dynamin inactivation on membrane morphology.**

A–F   TEM tomograms and 3D reconstructions of terminal cells in *shibire*$^{ts}$ embryos. (A) Control. (B-F) Kept at 34°C for 1 (B, E) or for 2 h (C, D, F). Boxed regions in (A) and (B) are magnified in (A″) and (B″). (A′, B′) Reconstructed 3D models. The tube membrane (cyan) is characterized by the electron-dense apical extracellular matrix (aECM). Invaginations with connections that can be traced to the tube membrane are red, and invaginations from the basal plasma membrane are green. The basal membrane is light pink. (C-D′) *shibire*$^{ts}$ terminal cell after 2 h at 34°C. (C′) 3D reconstruction, colour coding for basal and tube membrane is the same as above. Membrane that is continuous with both the apical and the basal membrane is shown in green, purple and dark blue. (D-D′) Higher magnification of a region in the same cell at a level where membrane sheets (purple and dark blue) bridge the apical and basal plasma membrane domains. (E-E″) Two focal planes and model of a tomogram from the cell shown in (B) where a basal (pink arrowhead) and an apical invagination (blue) are seen in close proximity (closest distance is marked by cyan and blue arrowheads). (F-F′) Apical membrane overgrowth regions of the cell shown in (C). The genotype of the embryos was *shibire*$^{ts}$; *btl* > KDEL:: RFP, Par3::YFP. The cell shown in (C–D) was found and acquired without the CLEM approach.

Data information: Scale bars: 500 nm.

overexpression phenocopies *shrb* loss of function (Michelet *et al*, 2010; Dong *et al*, 2014a).

Terminal cells from *shrb* mutant embryos had a morphology similar to that caused by dynamin inactivation: the basal membrane failed to grow properly, and large accumulations of the CD4 membrane reporter were seen within the cells (Fig 8I). Shrb::GFP overexpression in wild-type cells resulted in a similar, stronger phenotype (Fig EV5G and H). However, unlike in *shibire*$^{ts}$ cells, the aggregates that appear upon *shrb* loss of function did not contain PH::mCherry, indicating that they were not composed of plasma membrane (Fig 8J). Instead, the CD4::mIFP-marked membrane accumulation also carried the late endosomal marker Spin::RFP (Fig 8K). Taken together, this shows that the membrane accumulated inside the cell does not correspond to an overgrown apical compartment, but rather represents membrane material that was *en route* to MVBs, but, because of impaired Shrb function, was not incorporated into MVBs or processed further.

The finding that in this case, neither the basal plasma membrane, nor the apical membrane showed any growth indicates that both depend on membrane trafficking through MVBs. For the basal membrane, this fits with the observation that its growth depends on endocytosis, but for apical membrane, this was unexpected. It can mean either of two things. All newly synthesized membrane could be directed through the MVB to the apical domain, and the apical domain therefore does not grow if MVBs are disrupted because it never receives membrane. Alternatively, membrane could be passed to the apical domain from the Golgi or the ER directly, but is completely recycled through endocytosis and passage through the MVB. We suggest that only the second scenario is consistent with our observations in *shibire*$^{ts}$ cells: the MVBs, as well as other vesicles carrying late endosomal markers, disappear in these cells, and yet the apical membrane continues to grow. This shows that the initial delivery of membrane to the apical domain does not require the MVBs. We also confirmed this by blocking endocytosis in cells overexpressing Shrb::GFP, where we found that material still accumulated in the apical domain even though MVBs were no longer functional (Fig EV5I and J).

These results show that in terminal cells, apical-to-basal transcytosis goes through MVBs, and we wondered if this was also true for other transcytotic routes. Serpentine (Serp) is a chitin deacetylase located in the tracheal lumen. It is produced in and secreted by tracheal cells early during embryogenesis, but at later stages, it is produced in the fat body and transported to the tracheal lumen in a process involving basal-to-apical transcytosis (Dong *et al*, 2014b). Serpentine is also endocytosed from and recycled back to the tracheal lumen (Dong *et al*, 2013). To test whether these processes

depend on MVBs, we expressed Shrb::GFP in the tracheal system and looked at Serp distribution in the tracheal dorsal trunk. We found that in control embryos, Serp was only seen in the lumen (Fig 8L). In embryos expressing Shrb::GFP, Serp was also visible in cytoplasmic vesicles that were surrounded by Shrb::GFP itself (Fig 8M). This is consistent with results showing that *shrb* mutants accumulate Serp in tracheal cells (Dong *et al*, 2013). It is likely that Serp accumulation around Shrb::GFP is the result of basal-to-apical transcytosis and of apical-to-apical recycling of Serp in the dorsal trunk cells. Thus late endosomes also here appear to serve as stations that collect material from and re-deliver it towards various compartments of the cell.

## Discussion

Our results suggest a route taken by plasma membrane material during the coordinated growth of the apical and basal domains of tracheal terminal cells, summarized in the synopsis image. We will discuss the evidence that leads us to postulate this path, and how it relates to previously discovered functions for components of vesicle trafficking systems and MVBs in cell morphogenesis and transcytosis.

### Initial delivery of membrane

We suggest that apical plasma membrane comes from the ER and Golgi, as also shown in MDCK cells in 3D cultures (Ferrari *et al*, 2008). In tracheal terminal cells, membrane is delivered throughout the length of the tube rather than any specific region (Gervais & Casanova, 2010). This fits with ER and Golgi being distributed along the entire length of the branch. We show that blocking MVB biogenesis prevents membrane delivery to either membrane compartment, which could argue for a role for MVBs in initial membrane delivery. However, we also show that MVB function is downstream of endocytosis.

### Retrieval by endocytosis

Endocytosis plays important roles at multiple steps of tracheal development (Tsarouhas *et al*, 2007; Schottenfeld-Roames *et al*, 2014; Skouloudaki *et al*, 2019), but the fate of the membrane material ingested by the cell through apical endocytosis, the route it takes, and whether it contributes to elaborating the shape of the terminal cell were not known. In larval terminal cells, loss or reduction of endocytic components such as dynamin, Rab5 or syntaxin-7

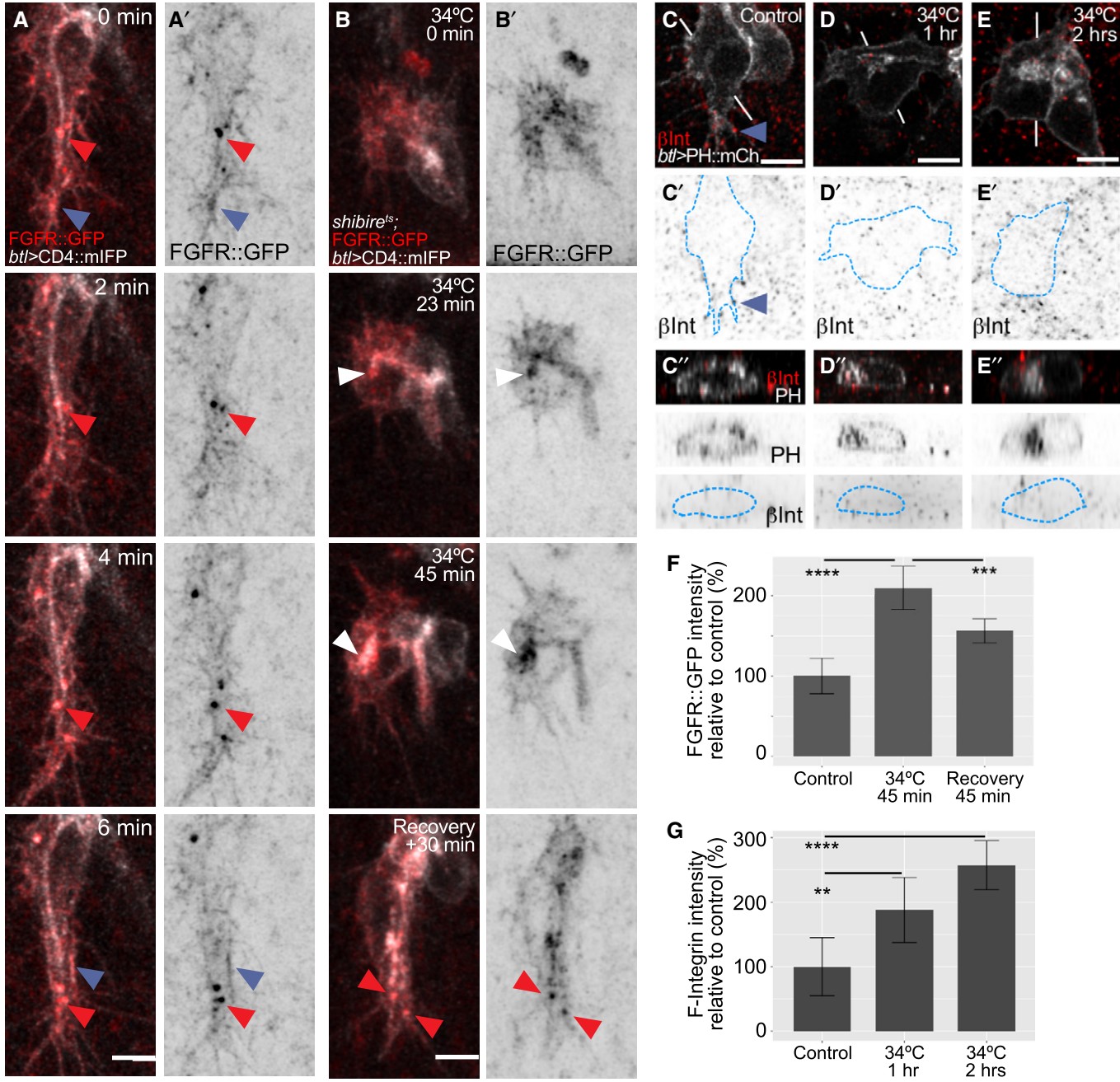

**Figure 6. Effect of dynamin inactivation on the distribution of basal proteins.**

A, B   Terminal cells expressing the membrane marker CD4::mIFP under *btl-gal4* and FGFR::GFP under its own promoter (from the fTRG library). (A) Time lapse imaging of a control cell. Blue arrowheads point to filopodia and basal plasma membrane, and red ones point to puncta containing CD4::mIFP and FGFR::GFP. (B) *shibire^ts* cell imaged before dynamin inactivation, after 23 and 45 min of inactivation, and after 30 min of recovery. White arrowheads point to FGFR::GFP accumulation at the apical compartment.

C–E   Single confocal planes of terminal cells expressing PH::mCherry and stained for βPS-integrin. The outline of the cells was traced using the PH::mCherry signal and is shown as a blue dashed line. (C) Control. Blue arrowheads: βPS-integrin signal in filopodia. (D) 1 h at 34°C; (E) 2 h at 34°C. (C″–E″) Orthogonal views of the lines shown in white in (C–E).

F, G   Quantification of fluorescence intensity from stained embryos of FGFR::GFP (F) and of βPS-integrin (G). Data are plotted as mean ± SD; significance was assessed using one-way ANOVA with Dunnett's correction for multiple comparisons. **$P$ = 0.0022, ***$P$ = 0.0003, ****$P$ < 0.0001. Number of cells analysed for (F): control, $n$ = 8; 1 h at 34°C, $n$ = 6; recovery, $n$ = 8. (G). For (G): control, $n$ = 7; 1 h at 34°C, $n$ = 8; 2 h at 34°C, $n$ = 8.

Data information: Scale bars: 5 μm.
Source data are available online for this figure.

results in cysts in the apical membrane and sometimes in extreme curling of the apical membrane within branches. These cells also form fewer branches per cell (Schottenfeld-Roames *et al*, 2014); thus, the phenotype suggests an increase in apical membrane at the

expense of basal membrane. The presence of excess apical membrane correlated with raised levels of Crb and depended on Crb. In our experiments, even though Crb aggregated at the tube membrane, knocking it down did not prevent apical membrane

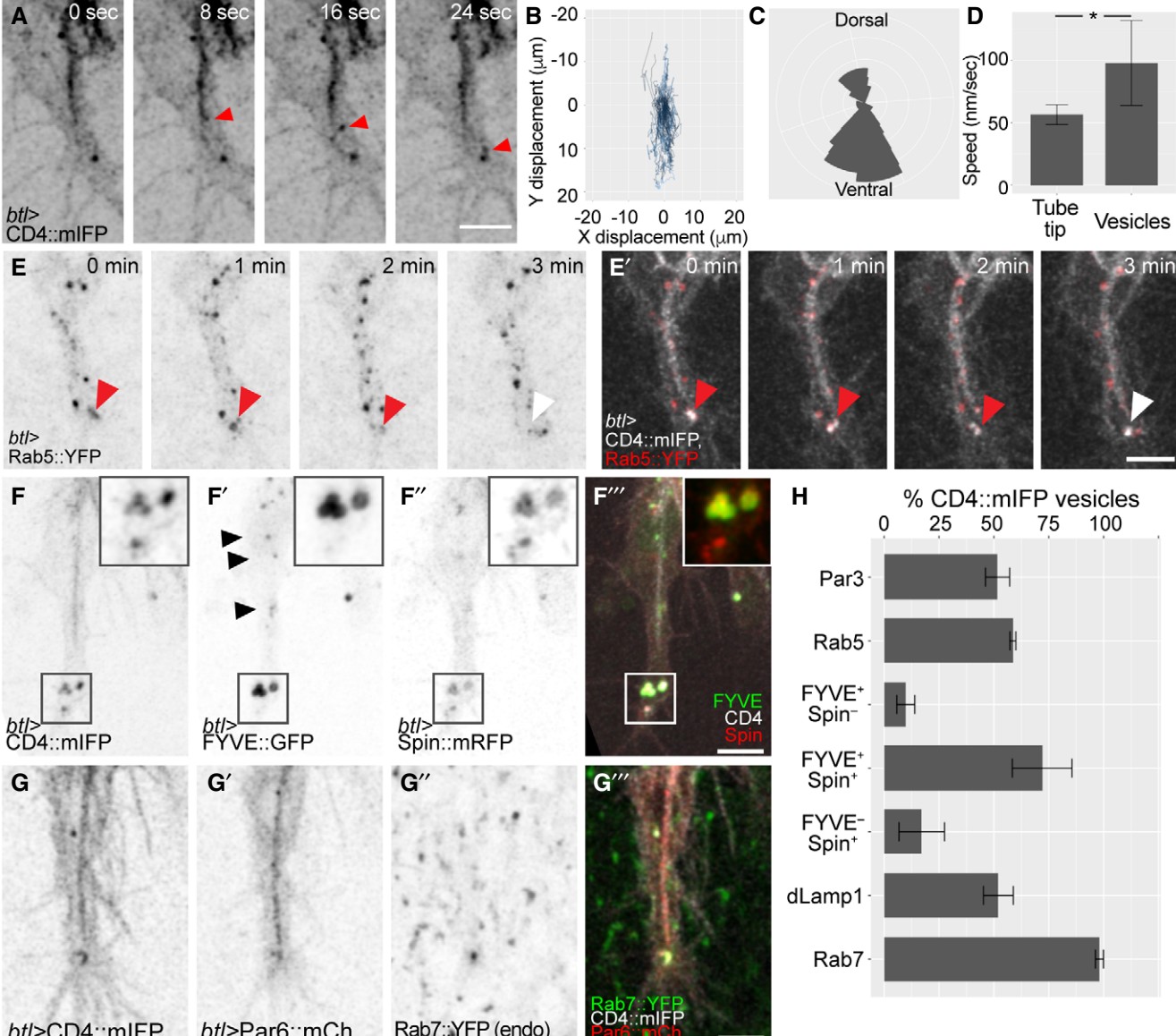

**Figure 7. Composition and distribution of endosomal compartments during terminal cell growth.**

A–D  High time resolution imaging of the membrane marker CD4::mIFP expressed under *btl-gal4*. (A) Example of a large CD4 vesicle (arrowhead) rapidly moving towards the tip of the terminal cell. (B–D) Analysis of 28 CD4 vesicles in five terminal cells. (B) Trajectories of CD4 vesicles with their original position mapped to the origin of the plot. (C) Rose diagram of the trajectories shown in (B). (D) Speed of CD4 vesicles compared to speed of tube growth measured over 25 min. Data are plotted as mean ± SD. *$P < 0.001$, Mann–Whitney U-test.

E–H  Terminal cells expressing the membrane reporter CD4::mIFP together with markers for vesicles of the endocytic pathway. (E) Rab5::YFP, red arrowheads point to a Rab5-positive vesicle; the white arrowhead shows the same vesicle once it lost the Rab5 signal. (F) FYVE::GFP, a PI$_3$P reporter; and Spin::mRFP, a lysosomal marker. Black arrowheads point to small FYVE::GFP vesicles, and the squared area shows large ones at the tip. (G) Par6::mCherry and endogenously labelled Rab7::YFP. (H) Percentage of CD4::mIFP vesicles that carry the indicated markers. Data are plotted as mean ± SD. For each marker, we analysed at least 20 min of cell growth with 10–20 time points. Number of cells analysed: Par3, $n = 2$; Rab5, $n = 3$; FYVE::GFP, $n = 4$; FYVE::GFP-Spin::RFP (which includes single- and double-positive vesicles), $n = 3$; dLamp1, $n = 2$; Rab7, $n = 5$.

Data information: Scale bars: 5 μm.
Source data are available online for this figure.

accumulation. Furthermore, upon recovery of dynamin function, Crb levels were similar to controls. We conclude that here, Crb accumulation does not account for the apical increase resulting from

endocytosis being blocked. The difference between the results in the embryo and the larva might be due to extremely different time-scales: while we acutely blocked endocytosis for periods of 15 min

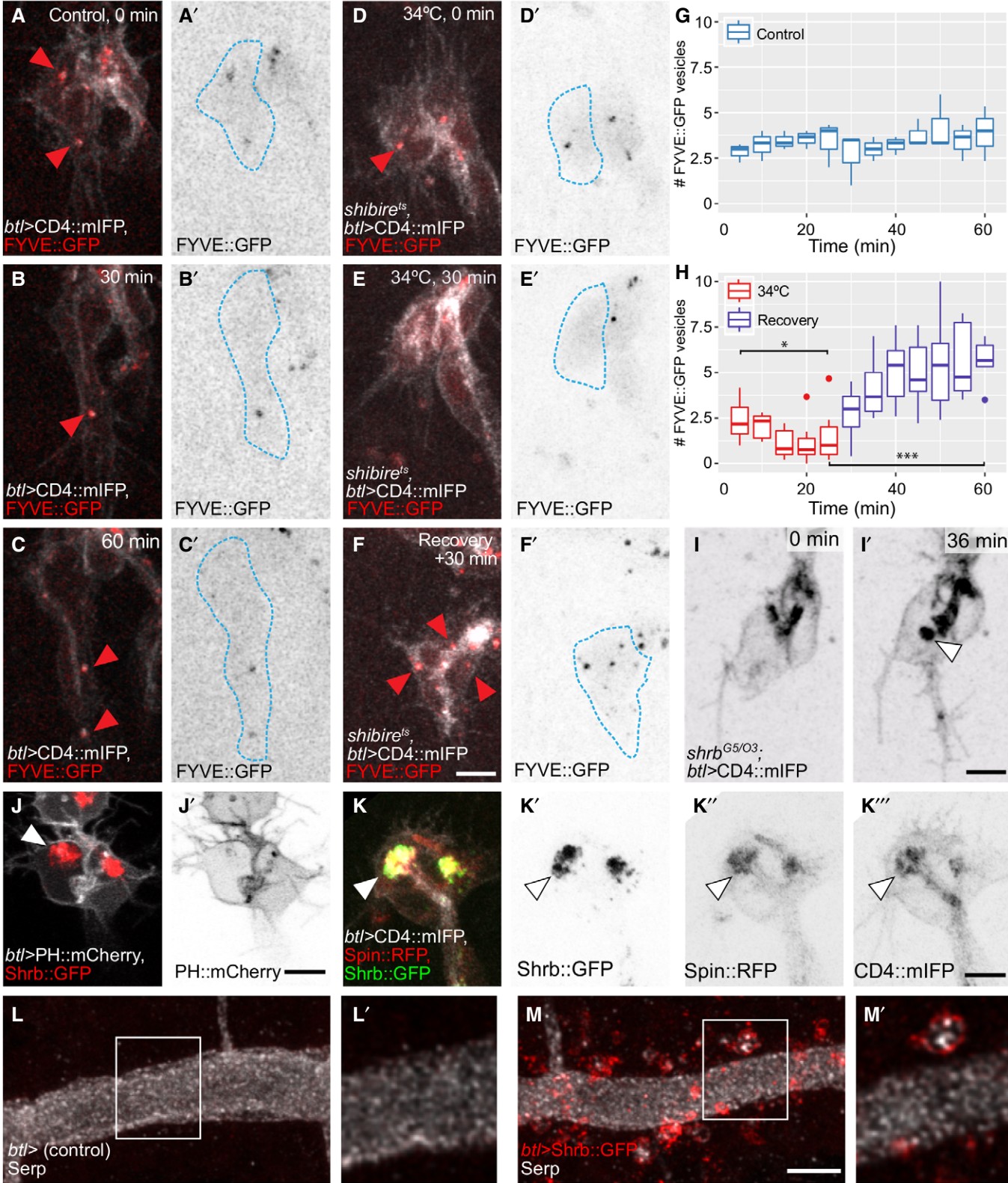

Figure 8.

**Figure 8. Distribution of late endosomal markers upon disruption of the endocytic pathway.**

A–F　Terminal cells expressing CD4::mIFP and FYVE::GFP under *btl-gal4*. The outline of the cells was traced using the CD4::mIFP signal and is shown as a blue dashed line. (A–C) Control cell. (D–F) *shibire*[ts] cell before 34°C (D), after 30 min at 34°C (E) and after 30 min of recovery (F). Red arrowheads: FYVE::GFP puncta.

G–H　Number of FYVE::GFP vesicles in 4 control cells (G) and in 6 *shibire*[ts] cells (H). Box plots represent median, interquartile range (IQR) and IQR*1.5 below and above the IQR. Significance in (H) was assessed using one-way ANOVA with Geisser–Greenhouse correction for paired data and Dunnett's multiple comparison test. *$P = 0.0288$, ***$P = 0.0016$.

I　　*shrb*[O3/G5] mutant cell expressing CD4::mIFP under *btl-gal4*. Arrowhead: CD4::mIFP aggregation.

J, K　Terminal cells overexpressing Shrb::GFP under *btl-gal4* together with PH::mCherry (J) or together with Spin::RFP and CD4::mIFP (K). Arrowheads: Shrb::GFP accumulations.

L, M　Dorsal trunk cells stained for Serp. In control embryos (L), Serp is seen exclusively in the tracheal lumen. The cells themselves are not visible in these images. In embryos expressing Shrb::GFP under *btl-gal4* (M), Serp is also seen inside the cells, usually in association with or surrounded by Shrb::GFP. Boxed regions are magnified in (L'-M') and shown as single confocal planes.

Data information: Scale bars: 5 μm.
Source data are available online for this figure.

---

to 2 h, larval cells that have grown for several days with reduced levels of endocytic components may be affected by compensatory processes, homeostatic regulation and longer-term interaction between membrane delivery and Crb accumulation.

## Transcytosis

Transcytosis has been documented in many cell types, most commonly in the transport of proteins and other cargo through epithelial barriers, which can occur either from basal to apical, or vice versa (Sasaki *et al*, 2007; Gallet *et al*, 2008; Callejo *et al*, 2011; Yamazaki *et al*, 2016). The classical example is immunoglobulin secretion and uptake. In MDCK cells, transcytosis of immunoglobulins occurs also in both directions and relies on Rab17 and Rab25. However, similar gene duplications and diversifications have not occurred in *Drosophila* (Gramates *et al*, 2017; Fung *et al*, 2018).

Our conclusion that membrane material is transcytosed from the apical to the basal membrane domain rests on two observations: in the absence of endocytosis, the apical membrane continues to grow but the basal stops, and large-scale physical connections appear between the basal and apical membrane after longer periods of blocking endocytosis. We view the sheets of membrane bridging the apical and basal membrane as evidence of membrane exchange between the two domains, representing structures that failed to be resolved normally in the absence of dynamin function. By this interpretation, unscissioned membrane invaginations protruding from the apical compartment, which are stripped of a clathrin coat, would occasionally have touched the basal plasma membrane or its protrusions and fused with it, as transcytosing vesicles would have done in the normal situation. Further plasma membrane delivery may then have expanded such initial channels into larger sheets. These membrane sheets also contain small vesicles similar to the ones produced by the ESCRT pathway to form MVBs, further supporting the delivery route we propose. Consistent with this model, severe dynamin inactivation in larval motor neurons results in a similar phenotype, with membrane cisternae bridging different regions of the synaptic bouton (Kasprowicz *et al*, 2014).

## Transcytosis as a morphogenetic mechanism

We are aware of only two cases where transcytosis is used as a morphogenetic mechanism: cellularization of the *Drosophila* blastoderm and morphogenesis of the excretory system in *C. elegans*

(Pelissier *et al*, 2003; Soulavie *et al*, 2018). During cellularization, endocytosis retrieves membrane material from the highly folded apical plasma membrane which is subsequently delivered to the leading edge of the growing lateral membranes (Pelissier *et al*, 2003). The *C. elegans* duct cell of the excretory system forms a subcellular tube by wrapping and auto-fusion of the plasma membrane. Basal-to-apical transcytosis is required for the elongation of this subcellular tube. This process does not depend on dynamin or clathrin but on AFF-1, a protein that also mediates the auto-fusion step (Sapir *et al*, 2007; Stone *et al*, 2009; Soulavie *et al*, 2018). Therefore, it is possible that the excretory duct cell adapted a different strategy to deliver membrane to build its subcellular tube, one that requires the AFF-1 fusogen instead of clathrin/dynamin-mediated endocytosis.

## MVBs in sorting and membrane delivery

We conclude that trafficking of membrane in the terminal cell goes through MVBs based on our functional results and the colocalization of a number of organelle, membrane and cargo markers. MVBs are known for their role in protein degradation, especially in the context of ligand-bound receptors (Michelet *et al*, 2010). However, recycling membrane with its associated transmembrane proteins back to its site of origin forms part of that process, and some observations can be interpreted as transcytosing material being associated with MVBs. Gold-labelled IgGs have been traced through various endosomal compartments, including MVBs, before they reach the basal membrane (He *et al*, 2008). Therefore, it is possible that newly synthesized proteins like FGFR reach MVBs but are then sorted out towards the basal compartment, while others like Crb are translocated into intraluminal vesicles and degraded, if not recycled back to the apical membrane. Observations in the dorsal trunk and in larval tracheal cells also indicate that this might be the case. Dorsal trunk tracheal cells lacking Shrb also accumulate Crb in late endosomes (Dong *et al*, 2014a). In larval terminal cells, loss of ESCRT-0 leads to intracellular accumulation of FGFR and also to reduced FGF signalling (Chanut-Delalande *et al*, 2010). According to our model, these results suggest that reduced FGF signalling activity in these cells is due to less FGFR reaching the basal plasma membrane. Thus, the defect is upstream of ligand–receptor interactions, rather than in signal transmission after receptor activation.

A multilayered membrane-bound compartment seen in immature larval tracheal branches (Nikolova & Metzstein, 2015) has not been

characterized in terms of marker distribution, but it is likely that it is functionally equivalent to the late endosomal compartments that we find at the growing tip of the embryonic terminal cell. Consistent with this, knockdown of vATPase components in larvae leads to formation of large intracellular Crb vesicles that are Rab5-positive, enlarged Lamp1-positive compartments and a distorted tube morphology (Francis & Ghabrial, 2015). Cells of the tracheal system that undergo anastomosis rely on specialized late endocytic vesicles called secretory lysosomes to drive tube fusion (Caviglia *et al*, 2016). These compartments are also positive for Rab7, and their secretion contributes to building a lumen between the fusing cells. MVBs may also be involved in the biogenesis of these vesicles; however, downstream mechanisms are likely to differ from the ones employed by terminal cells since fusion cells express a subset of proteins that have no role in terminal cell morphogenesis like Rab39 and the C2 domain protein Staccato (Caviglia *et al*, 2016).

In summary, our results show that plasma membrane turnover through the late endocytic pathway is a morphogenetic mechanism in which MVBs act as a hub for membrane and cargo sorting. This mechanism entails a massive plasma membrane flow that had so far not been documented. A role for MVBs in membrane homeostasis may be relevant for other cells with complex shapes, like neurons, which also rely heavily on polarized membrane trafficking to build long projections and to communicate with other tissues. While axons are generally devoid of MVBs, axon termini and dendrites frequently contain MVBs. MVBs have been proposed to serve as local stations to allow recycling, degradation or even storage of membrane material at a distance from the cell nucleus (Von Bartheld & Altick, 2011). Terminal cells, whose complexity increases dramatically within a few hours in terms of size and morphology, may represent an extreme case of membrane remodelling that relies on this mechanism, but this may nevertheless be required in the building of other complex cell shapes.

# Materials and Methods

## Fly lines

*UAS-PLCδ-PH-Cherry* was generated by standard molecular biology techniques using UAS-*PLCδ-PH-GFP* as template and cloned into pUAST-attB. The construct was inserted in the VK00033 locus. *btl-gal4* was used to drive UAS transgene expression in the trachea and was obtained from Markus Affolter laboratory, University of Basel, Switzerland (3rd chromosome), and Kyoto *Drosophila* Genetic Resource Center (#109128, 2nd chromosome). The following lines are from Bloomington: *UAS-SrcGFP* (#5432), *UAS-GalT-GFP* (#30902), *UAS-RFP-KDEL* (#30909), *UAS-IVS-myr::tdTomato* (#32221), *UAS-shrbGFP* (#32559), *UAS-GFP-myc-2xFYVE* and *UAS-Spin::mRFP* (#42716), and *UAS-CD4::mIFP-T2A-HO1* (#64182, #64183). From VDRC, we obtained *FGFR::GFP* (#318302) and *UAS-crb-IR* (#39177). We are grateful to the groups that kindly shared the following lines: endogenously labelled Rab7 and Rab11 (*YRab7* and *YRab11*), and Crb::GFP from Marko Brankatschk, TU Dresden, Germany (Dunst *et al*, 2015); *UAS-CLC::GFP, UAS-Shibire::YFP, UAS-Rab5::YFP, UAS-Rab11::GFP* and *shibire^ts1* from Stefano De Renzis, EMBL, Germany (Fabrowski *et al*, 2013); *UAS-3xeYFP-Baz* (Par3::YFP) from Chris Doe, University of Oregon, USA (Siller *et al*,

2006); *tub-Rab5::CFP* from Suzanne Eaton, MPI-CBG, Germany (Marois *et al*, 2006); *dLamp::mCherry* from Gabor Juhasz, Eotvos Lorand University, Hungary (Hegedus *et al*, 2016); UAS-*crb^extraTM::GFP* from Elisabeth Knust, MPI-CBG, Germany (Johnson *et al*, 2002); *UAS-PLCδ-PH::GFP* from Thomas Lecuit, IBDM, France (JayaNandanan *et al*, 2014); *UAS-palm::NeonGreen* from Stefan Luschnig, University of Münster, Germany (Sauerwald *et al*, 2017); *UAS-ANF::GFP* and *UAS-GASP::GFP* from Christos Samakovlis, Stockholm University, Sweden (Tsarouhas *et al*, 2007; Tiklova *et al*, 2013); and *UAS-Par6::mCherry* from Daniel St Johnston, The Gurdon Institute, UK (Doerflinger *et al*, 2010). We used *shrb* null alleles from Bloomington, *shrb^O3* (#39623) and *shrb^G5* (#39635). We balanced them over *CyO{dfd-YFP}* and added *btl-gal4, UAS-CD4::mIFP* on the 3rd chromosome. *shrb^O3/shrb^G5* embryos were selected by the absence of *dfd*-YFP.

## Live imaging

Embryos were dechorionated in a 50% bleach solution, washed and mounted in halocarbon oil using glass coverslips or MatTek glass bottom dishes and heptane glue. Experiments at 34°C were done using a custom incubator chamber mounted on the microscope. For recovery experiments, embryos were mounted in water instead of oil, incubated at 34°C for the desired time, and for recovery, temperature was lowered to 23°C and ice-cold water was added. Samples were imaged using a Zeiss LSM 880 in Airyscan Fast Mode, Perkin-Elmer UltraVIEW VoX or UltraVIEW ERS spinning disc confocal microscopes using Plan-Apochromat 63×/1.4 Oil objectives. Airyscan images were deconvolved using the Zeiss ZEN software using the auto settings. Spinning disc images were deconvolved using Huygens Professional, SVI, and processed in FIJI. Unless otherwise stated, images are presented as maximum-intensity z-projections.

## Immunostainings

Embryos were dechorionated, devitellinized and fixed using 37% paraformaldehyde for 15 s while vortexing and 5 min in a rocker. Afterwards, embryos were blocked using bovine serum albumin and incubated overnight at 4°C in primary antibody solution, and secondary antibodies were incubated for 2 h the following day. Embryos were mounted in VECTASHIELD. We used the following antibodies: mouse anti-βPS Integrin (1:200, DSHB #6G11), rat anti-Crb (1:500, gift from Elisabeth Knust, MPI-CBG, Germany), rat anti-ECad (1:100, DSHB #DCAD2), mouse anti-FasIII (1:200 DSHB #7G10), mouse anti-dpERK (1:200, Sigma-Aldrich #M8159), rabbit anti-Dof [1:200 (Vincent *et al*, 1998)] and rabbit anti-Serp (1:300, gift from Stefan Luschnig, University of Münster, Germany). CBD conjugated to Alexa 647 was described previously (JayaNandanan *et al*, 2014). To enhance signal from fluorescent reporters, we used GFP-booster coupled to Atto488 (gba488) and RFP-booster coupled to Atto594 (rba594) from Chromotek. Secondary antibodies used were from Thermo Scientific: Alexa 568 goat anti-mouse (1:500, A-11031), Alexa 568 goat anti-rat (1:500, A-11077) or Alexa 647 goat anti-rabbit (1:300, A-21245). Embryos were imaged using a Zeiss LSM 780 using a Plan-Apochromat 63×/1.4 Oil objective and deconvolved with Huygens Professional and processed in

FIJI. Unless otherwise stated, images are presented as maximum-intensity z-projections.

## Electron tomography

### *Initial EM workflow*

The embryos were staged and high-pressure frozen (HPM010 Abra Fluid) in 20% Ficoll (mol weight ~70,000). The freeze-substitution was done (EM-AFS2, Leica Microsystems) with 0.3% uranyl acetate, 0.3% glutaraldehyde and 3% water in acetone at −90°C for 48 h. The temperature was then raised to −45°C at 3.5°C/h, and samples were further incubated for 5 h. The samples were rinsed in acetone, followed by infiltration in Lowicryl HM20 resin, while raising the temperature to −25°C. The samples were polymerized under UV light for 48 h at −25°C and for further 9 h, while the temperature was gradually raised to 20°C (5°C/h). Thick sections (300 nm) were cut from the polymerized resin block and picked up on carbon and Formvar-coated slot grids.

The sections were screened using a FEI Tecnai F30 electron microscope with a Gatan OneView camera and acquiring large field-of-view montages with SerialEM (summarized in Appendix Fig S2). The serial section montages were aligned and segmented using the IMOD package (Mastronarde, 1997) in order to find terminal cells. The sections that had terminal cells were then imaged again for electron tomography.

### *CLEM workflow*

High-pressure freezing and freeze-substitution were done as above, except that freeze-substitution was done with 0.1% uranyl acetate in acetone at −90°C for 48 h. The fluorescence microscopy imaging of the sections was carried out as previously described (Hampoelz *et al*, 2016) using a widefield fluorescence microscope (Olympus IX81). The images collected were used to screen for the sections with terminal cells. Those sections were then used for electron tomography. For electron tomography, tilt series were performed in 1 degree increments from 60 to −60 degrees with 2.549 nm or 0.78 nm pixel size on a FEI Tecnai F30 electron microscope with a Gatan OneView camera. The serial tomograms were reconstructed, aligned and segmented using the IMOD package.

## Image analyses

Tube length and vesicle displacement were analysed in FIJI using the manual tracking plug-in. Organelle distance from the base of the cell was measured manually in FIJI, setting the junction to the previous cell as origin and measuring the distance along the tube to the tip of the cell.

To determine the total amount of membrane present in the cells, we did z SUM projections and then determined background signal outside the cell and subtracted this from the pixel values in the SUM. We manually segmented the cell and the tube within the cell. To determine the fluorescence intensity of the apical membrane, we measured the total pixel intensity values (RawIntDen) within the apical membrane. The z SUM in this area also includes signal from the basal membrane below and above the apical tube. We determined this amount by measuring an area next to the tube and subtracted this value from the one we had measured for the tube. We took the resulting value to be the membrane in the apical domain. To determine the amount of membrane in the basal domain, we therefore simply subtracted the apical value from the total value.

To quantify the amount of βPS-integrin at the apical membrane compartment, we subtracted background, manually segmented the tube within the cell from z SUM substacks using FIJI and quantified the pixel values for βPS-integrin within it. The values were normalized to the mean pixel intensity values surrounding the cell. For Crb, we used a similar approach but instead we measured the pixel values at the apical membrane of the dorsal trunk cells and normalized them to the signal to the values of the epidermis. Marker occupancy at CD4 vesicles and number of vesicles in time were scored manually in FIJI.

FGFR and dpERK signal quantifications were done by volume in Imaris, using the Dof signal as 3D mask. Since Dof is a cytoplasmic protein and it does not label filopodia, using it for segmentation allowed us to discard the amount of FGFR still present in the basal plasma membrane.

EM images were analysed using the IMOD package. The number of vesicles on a given tomogram was normalized to the area covered by cytoplasm at the mid z-plane of the tomogram. Large vesicles like MVBs can span more than one tomogram, and we ensured these were only counted once whenever they were visible in more than one tomogram. Endocytic events were normalized to the surface of plasma membrane visible on a given tomogram.

## Statistical analyses

We used GraphPad Prism 6 for all statistical analyses. Plots were generated using RStudio and the ggplot2 package.

# Data availability

The electron microscopy tomograms presented in Figs 2, 3 and 5 have been deposited at the Electron Microscopy Public Image Archive (EMPIAR, https://www.ebi.ac.uk/pdbe/emdb/empiar/) with the accession number EMPIAR-10434.

**Expanded View** for this article is available online.

## Acknowledgements

We thank Stefano De Renzis, Sandra Iden, Natalia Kononenko, Blanche Schwappach, Catherine Rabouille and members of the Leptin laboratory for helpful discussions; the Vienna *Drosophila* Resource Center, Bloomington *Drosophila* Stock Center, Kyoto *Drosophila* Genetic Resource Center and our colleagues Markus Affolter, Marko Brankatschk, Stefano De Renzis, Chris Doe, Suzanne Eaton, Gabor Juhasz, Elisabeth Knust, Thomas Lecuit, Stefan Luschnig, Christos Samakovlis and Daniel St Johnson for stocks and reagents; the EMBL Advanced Light Microscopy Facility (ALMF) for continuous support; Zeiss for the support of the ALMF; and FlyBase. This work was supported by funding from EMBL and EMBO. LDRB was funded by the EMBL Interdisciplinary Post-doctoral Programme under Marie Curie Actions. We thank Eduardo Rojas-Hortelano for the EM tomogram tracing and analyses. Open access funding enabled and organized by Projekt DEAL.

## Author contributions

MR, R-BLD and LM conceived the study. MR, R-BLD and MP contributed to methodology and investigation. R-BLD performed formal analysis and wrote

the original draft of the manuscript. MR, R-BLD, MP, SY and LM wrote, reviewed and edited the manuscript. R-BLD contributed to visualization. SY and LM supervised the study. LM acquired funding.

## Conflict of interest

Maria Leptin is director of EMBO. *The EMBO Journal* is editorially independent of EMBO.

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
