## [Review Process File · The EMBO Journal]

Transcytosis via the late endocytic pathway as a cell morphogenetic mechanism

Renjith Mathew, Luis Rios-Barrera, Pedro Machado, Yannick Schwab, and Maria Leptin

DOI: [10.15252/emboj.2020105332](https://doi.org/10.15252/emboj.2020105332)

Corresponding authors: Maria Leptin (leptin@embo.org)

Review Timeline:

Transfer from Review Commons:	18th Apr 20
Editorial Decision:	27th Apr 20
Authors' Correspondence:	2 th Apr 20
Authors' Correspondence:	30th Apr 20
Editor's Correspondence:	30th Apr 20
Revision Received:	18th May 20
Editorial Decision:	9th Jun 20
Revision Received:	10th Jun 20
Accepted:	16th Jun 20

Editor: Ieva Gailite

Transaction Report: This manuscript was transferred to the EMBO Journal following peer review at Review Commons.

Review
COMMONS

Thank you for submitting your manuscript for consideration by the EMBO Journal. I sincerely apologise for the delay in the assessment of your study due to the high number of submissions our office is experiencing at the moment and protracted discussions with my colleagues.

I have now read your manuscript, the reviewer comments and your revision proposal. Given the referees' positive assessment, I would like to invite you to submit a revised version of the manuscript, addressing the comments of all three reviewers. We find the preliminary revision plan reasonable and appreciate the limitations imposed by the current COVID-19/SARS-CoV-2 pandemic. From our side, we found the additional point raised by reviewer #1 regarding apical/basal cargo co-localisation in multi-vesicular bodies important, and we would like to offer an extended timeline of revision to allow inclusion of this data. I would be happy to discuss the revision in more detail via email or phone/videoconferencing.

Due to the delays to experimental work caused by the ongoing pandemic, we have extended our 'scooping protection policy' beyond the usual 3 month revision timeline to cover the period required for a full revision to address the essential experimental issues. This means that competing manuscripts published during revision period will not negatively impact on our assessment of the conceptual advance presented by your study. Please contact me if you see a paper with related content published elsewhere to discuss the appropriate course of action.

Please feel free to contact me if you have any further questions regarding the revision. Thank you for the opportunity to consider your work for publication. I look forward to receiving your revised manuscript.

Reviewer #1 (Evidence, reproducibility and clarity (Required)):

****Summary:****

In this work the authors investigate the dynamics of cell morphogenesis in a convenient in vivo system. They use the terminal cells of the embryonic tracheal system and comprehensively address how cell shape change (elongation in this case) takes place and how membrane is remodeled during the process. By combining different high-resolution techniques, i.e. in vivo imaging of terminal cells expressing different membrane markers and serial-section electron tomography, they describe the organelle organization (ER, Golgi, different types of vesicles) in the terminal cells during their elongation. They identify the presence of membrane structures/vesicles particularly abundant at the tip of the cell ahead of the growing tube. When they block endocytosis they find increased tube membrane and lack of basal membrane growth. In addition, in tomograms, they observe clear membrane defects, like invaginations that could even connect the tube and basal membranes. This correlates with the absence of vesicles at the tip observed in normal conditions.

The analysis of the nature of the vesicles observed indicated the accumulation of late endosomes and MVB, particularly at the tip of the growing terminal cells. Interfering with the formation of these MVB led to defects in the growth of both the tube and the basal membrane.

Altogether the authors propose a model in which the newly formed membrane (and transmembrane proteins) passes through the ER and Golgi and reaches the apical membrane. The incorporated membrane is then rapidly endocytosed and follows a maturation pathway through MVB, from where different cargoes and membrane would be sorted and recycled back to the apical (tube) membrane, or to the basal membrane through a transcytosis mechanism

****Major comments:****

- Are the key conclusions convincing?

Most of the conclusions presented are convincing and supported by the results observed. However, to my understanding, one of the key conclusions of this work (that membrane is transcytosed from the apical to the basal domain) is not fully convincing. A critical result to support the author's conclusion of apical-basal transcytosis is to find clear evidence of basal accumulation of a transcytosed marker. The authors show accumulation of FGFR-GFP and Myospheroid as evidence. However, I find the results presented not very convincing. The accumulation of FGFR-GFP at the basal membrane in the control is not very clear in the images and movie presented. In addition, in the shibire mutant, some basal accumulation of FGFR-GFP seems to be detected (particularly in the movie). In the figures the authors show an increase of FGFR-GFP intensity when endocytosis is blocked, but this is not explained in the text.

On the other hand, and very importantly, how does FGFR localization relates to its activity? The authors show that when endocytosis is prevented dpERK (i.e. a reporter of FGFR activation) is not decreased, indicating that FGFR is normally active. Wouldn't this suggest that FGFR is still localized basally to receive Bnl signal? Actually, as the authors indicate, in

larval tracheal cells the intracellular accumulation of the FGFR leads to a reduced FGF signal transduction (Chanut-Delalande et al., 2010), suggesting that reduced FGF signaling activity in these cells is due to less FGFR reaching the basal membrane.

The results with Myospheroid are not very convincing either, as the authors just show a single confocal section of control and shibire mutants.

In summary, I consider that this very important point needs to be better documented before concluding that apical membrane material containing basal cargoes is transcytosed to the basal membrane.

Another conclusion that, to my opinion, should be better explained and documented, is the coordination of tube and basal membrane growth. Following the movements of the vesicles the authors conclude that there is a net displacement of these vesicles to the tip of the cell. This correlates with the presence of mature endosomes there. So the results postulate that transcytosis occurs at the tip, and therefore the growth of the basal membrane would occur preferentially at the tip. It has been demonstrated, as the authors indicate, that the tube membrane grows all along the length of the tube. How is then coordinated the tube and basal membrane growth? If, as the authors propose in their model, the tube membrane also grows after a process of endocytosis and recycling, wouldn't it be expected to have preferential tip growth? How do the authors reconcile all these observations with previously published results (Gervais and Casanova, 2010)?

The serial-section electron tomography analysis is very interesting and identifies different sorts of vesicles. However, it is very unclear what the different vesicles referred in the models correspond to (in the *in vivo* imaging for instance). For instance, the small granular or the dense-core vesicles correspond to endocytic vesicles at different stages of maturation?. If there is a constant endocytosis from the apical membrane to generate the basal and build the definitive apical membrane, wouldn't it be expected to find many more vesicles around the tube? Wouldn't it be expected to find coated vesicles around or budding from the tube, as the coated vesicles observed budding from the basal membrane in Fig 2D? Or is the endocytosis observed mediated by non-clathrin coated vesicles?.

- Should the authors qualify some of their claims as preliminary or speculative, or remove them altogether?

The results of Serp accumulation upon MVB interference can lead to confusion (from line 404). The authors seem to suggest that Serp protein is exclusively produced in the FB and transported by transcytosis to reach the tracheal lumen. However, Serp is also produced in the tracheal cells themselves. In fact, serp expression in the FB seems to be detected by late embryogenesis, while expression in tracheal cells is detected much earlier (Dong et al. 2014; Luschnig et al. 2006; Wang et al. 2006). It was also shown that Serp undergoes a recycling mechanism from the lumen to the lumen, through the endosomes-TGN retrograde trafficking, that may also require Shrub (Dong et al, 2014; Dong et al 2013). Thus, it is unclear (and even unlikely) that the Serp found in the vesicles in Shrub-GFP mutants is derived exclusively from the transcytosed component from the FB.

I suggest to better explore this issue or to remove this part.

- Would additional experiments be essential to support the claims of the paper? Request additional experiments only where necessary for the paper as it is, and do not ask authors to open new lines of experimentation.

As indicated before, more conclusive results for transcytosis should be provided.

I suggest that the authors determine the presence of apical and basal cargoes (FGFR) in the late endosomes found when Shrub activity is impaired. According to his model, both types of markers should accumulate there. A high accumulation of these markers in those endosomes would reinforce the hypothesis proposed.

While it has been reported that shrub-GFP act as a dominant negative in different contexts (Dong et al, 2014; Sweeney et al 2006) it is unclear why. So it would be desirable to confirm the results with a loss of function condition (either mutant or RNAi line)

- Are the data and the methods presented in such a way that they can be reproduced?

The materials and methods section would benefit from more detailed explanations.

- Are the experiments adequately replicated and statistical analysis adequate?

Many of the experiments presented in this work are technically very challenging, like the in vivo analyses and particularly the serial-section electron tomography. This prevents having high numbers of replicates on occasions.

****Minor comments:****

-In the abstract the authors state: "We show that apical endocytosis and late endosome-mediated trafficking determine the membrane allocation to the apical and basal membrane domains". I think that the authors show that "that apical endocytosis and late endosome-mediated trafficking is required for correct membrane growth", but I am not that sure that they show that it determines the membrane allocation

-References for the PH-GFP localization in cells should be provided. Which is the evidence that it only localizes to plasma membrane?

-It would be more adequate to always use the same terms to facilitate the reading. For instance, in several figures the membranes are referred as basal plasma membrane and tube membrane, but in others outer and inner membrane

-Figure 3C,D and corresponding text are difficult to understand. The increase of fluorescence of the inner membrane seems to be very high, even higher than the corresponding to the outer membrane. Can the authors explain better this point and also describe better the method applied in the materials and methods section?

Reviewer #1 (Significance (Required)):

This work represents an important advance for the field for several reasons. First of all it represents a technical advance because the authors are able to combine the traditional genetic analysis with two powerful techniques (in vivo imaging and serial-section electron tomography) to analyze single cell behavior at high resolution (temporal and spatial). In addition it represents a conceptual advance as it proposes a mechanism through which membrane growth is coordinated to regulate cell morphogenesis. The mechanism presented (endocytosis and transcytosis) is not new but they find evidence in an in vivo system.

It was previously known that tracheal terminal cells undergo a process of intracellular tube formation and cell elongation at the same time, but the mechanisms coordinating these two cell events were not known. The proposed mechanism may not only be relevant for the morphogenesis of tracheal terminal cells, but could represent a general mechanism of cell morphogenesis. Therefore, the paper should be relevant for research in the morphogenesis area but also in the cell biology field, as it shows how regulated membrane trafficking can control tissue morphogenesis

REFEREES CROSS-COMMENTING:

I agree with reviewer #4 on her/his comments and suggestions about analyzing the involvement of the recycling endosome in the process.

Reviewer #2 (Evidence, reproducibility and clarity (Required)):

In their manuscript, Mathew et al present a model in which transcytosis is utilized to deliver endocytosed apical membranes to supply basal membrane growth. The authors examined the developing terminal cell of the fruitfly tracheal system, which is a well established tubulogenesis model, as these cells form subcellular tubes by apical plasma membrane invagination. The authors show that basal membrane growth stops when endocytosis or endosomal transport is blocked, while the apical membrane grows excessively or membrane material accumulates in the cytosol, respectively.

Reviewer #2 (Significance (Required)):

The authors used high-end microscopy (including CLEM and electron tomography) to support their model and in my opinion, the quality and the quantity of the presented data are indeed adequate for this. The text is well written, the figures are of superb quality, and several cartoons help to understand the presented data/experiments. Therefore I highly recommend submitting the manuscript to a cell biology journal in its present form.

Reviewer #3 (Evidence, reproducibility and clarity (Required)):

This paper investigates the role of membrane trafficking in growth of the polarized tracheal cell that forms a cellular projection containing a subcellular tube. The authors show that apical endocytosis and late endosome-mediated trafficking determine the membrane allocation to the apical and basal membrane domains. Basal plasma membrane growth stops if endocytosis is blocked, whereas the apical membrane grows excessively. Plasma membrane is initially delivered apically, and then appears to be continuously endocytosed, together with apical and basal cargo. The sorting and recycling of apical and basolateral membrane appears to occur in a novel organelle carrying markers of late endosomes and multivesicular bodies (MVBs). Inhibiting endocytosis eliminates this compartment. The work in this paper is generally of high quality, and with one exception, quite comprehensive. The writing and figures are clear.

****Major concerns:****

-A central focus of the paper is that balance between apical and basal membrane and the role of transcytosis in moving membrane from the apical compartment to the basolateral compartment. The current view is that transcytosis in mammalian cells usually goes through the recycling endosomes which are marked by rab11, although there is evidence for some trafficking through MVBs as well. In *Drosophila*, Rab11 positive recycling endosomes are frequently examined as part of endocytic system analyses. However, rab11 is not used as a marker in this paper and indeed there is no mention at all of recycling endosomes, even though recycling is at the core of the work. Since the authors do not examine rab11 or other possible markers of recycling endosomes, it is unclear whether the organelle they identify as carrying markers of late endosome and MVBs is some MVB/recycling endosome hybrid, or whether the organelle is completely distinct from the recycling endosome. Consequently, it is not possible to assess whether the observed trafficking either uses or does not use involve the recycling endosome. This ambiguity makes it difficult to relate the observed trafficking to other systems. Minimally the authors should stain for rab-11 in WT and in some of the conditions where trafficking is perturbed and determine if the MVB-like compartment they are observing is rab11 positive, and whether the recycling endosome are affected by the perturbations. Further experiments may be needed to resolve whether any trafficking is going via the recycling endosome or this new MVB-type structure, but without even preliminary data on the relationship between the MVB compartment and the recycling endosome, it's hard to say what might be appropriate or exactly how long addressing this will take. But just staining for rab11 in WT and a few mutant conditions to get a handle on what is up with the recycling endosomes in these cells should take less than a month.

-In addition to the above, I would recommend more discussion of how the authors' results relate to membrane trafficking and transcytosis in other systems. The recycling endosome should be considered, and it may be appropriate to draw comparisons to membrane trafficking in neurons that goes through MVBs (e.g. reviewed in VON BARTHELD and ALTICK *Prog Neurobiol.* 2011 Mar; 93(3): 313-340.). Although neurons are not hollow, they have definite morphological resemblance to tracheal terminal branches.

-line 204-208 "To test whether raised levels of Crb were responsible for the excessive apical membrane, as reported in other contexts (Pellikka et al., 2002; Schottenfeld-Roames et al.,

2014; Wodarz et al., 1995), we knocked down Crb (Fig. S4G-H)." According to the legend for Fig s4, the authors express an RNAi construct against crbs. However, there does not appear to be any quantification of the amount knockdown of crb that was achieved. This is a concern for two reasons: 1) RNAi in the embryonic trachea works poorly for most genes (for unknown reasons) 2) This does not appear to be a clonal experiment but rather a pan tracheal driving of Crb RNAi. Loss of crbs would be expected to have very negative effects on tracheal morphogenesis (although this hasn't been rigorously tested to this reviewer's knowledge), but there doesn't appear to be any adverse effects of pan-tracheal crbs RNAi, suggesting that little if any knockdown of crb was actually achieved.

The authors either need to document the reduction of crbs or remove this paragraph. Preferably, they would be able to document the reduction of crb because they are trying to address an important point and if they can show the apical expansion is crb-independent, that would be a nice result.

****Minor concerns:****

-Lines 250. "By this interpretation, unscissioned membrane invaginations protruding from the subcellular tube would occasionally have touched the basal plasma membrane or its protrusions and fused with it, as transcytosing vesicles would have done in the normal situation."

I am not convinced by the argument that the bridging invaginations are fusing analogously to transcytosing vesicles because a protrusion/nascent vesicle coming from the apical surface should have rab5 and V-SNAREs that should dock the protrusion/nascent vesicle with an endosomal compartment, not the basolateral surface. A transcytotic vesicle would have the rab7 and V-SNAREs for the basolateral membrane. So it would seem that a fusion of the apical surface directly to the basolateral surface would have to be an ectopic event outside of the normal situation.

-Significant figures. This is not a big deal, but the authors are over reporting their significant figures. E.g. "a 7.05-fold increase (+/-2.98 SD)". With an SD that is 50% the value of the measurement, reporting to hundreds is definitely beyond the accuracy of measurement. Rounding to tenths would be more appropriate.

Reviewer #3 (Significance (Required)):

As there have not been that many studies on the dynamics of membrane trafficking during morphogenesis, the results should be of broad interest to those studying the endocytic system and the role of membrane trafficking in morphogenesis. However, the paper would be greatly strengthened if the authors considered the recycling endosome in their analysis and write up. As a well-known compartment for trafficking cargo and membrane to both the apical and basolateral surface, it is hard to know how to interpret the observed trafficking without knowing the involvement, or lack thereof, of recycling endosomes in this system.

Authors' revision proposal

Detailed response to referees (our responses in blue font)

Reviewer #1 (Evidence, reproducibility and clarity (Required)):

****Summary:****

In this work the authors investigate the dynamics of cell morphogenesis in a convenient in vivo system. They use the terminal cells of the embryonic tracheal system and comprehensively address how cell shape change (elongation in this case) takes place and how membrane is remodeled during the process. By combining different high-resolution techniques, i.e. in vivo imaging of terminal cells expressing different membrane markers and serial-section electron tomography, they describe the organelle organization (ER, Golgi, different types of vesicles) in the terminal cells during their elongation. They identify the presence of membrane structures/vesicles particularly abundant at the tip of the cell ahead of the growing tube. When they block endocytosis they find increased tube membrane and lack of basal membrane growth. In addition, in tomograms, they observe clear membrane defects, like invaginations that could even connect the tube and basal membranes. This correlates with the absence of vesicles at the tip observed in normal conditions.

The analysis of the nature of the vesicles observed indicated the accumulation of late endosomes and MVB, particularly at the tip of the growing terminal cells. Interfering with the formation of these MVB led to defects in the growth of both the tube and the basal membrane.

Altogether the authors propose a model in which the newly formed membrane (and transmembrane proteins) passes through the ER and Golgi and reaches the apical membrane. The incorporated membrane is then rapidly endocytosed and follows a maturation pathway through MVB, from where different cargoes and membrane would be sorted and recycled back to the apical (tube) membrane, or to the basal membrane through a transcytosis mechanism

****Major comments:****

- Are the key conclusions convincing?

Most of the conclusions presented are convincing and supported by the results observed.

However, to my understanding, one of the key conclusions of this work (that membrane is transcytosed from the apical to the basal domain) is not fully convincing. A critical result to support the author's conclusion of apical-basal transcytosis is to find clear evidence of basal accumulation of a transcytosed marker.

The authors show accumulation of FGFR-GFP and Myospheroid as evidence. However, I find the results presented not very convincing. The accumulation of FGFR-GFP at the basal membrane in the control is not very clear in the images and movie presented. In addition, in the shibire mutant, some basal accumulation of FGFR-GFP seems to be detected (particularly in the movie). In the figures the authors show an increase of FGFR-GFP intensity when endocytosis is blocked, but this is not explained in the text.

If we understand the referee correctly, there are two parts to her/his concern:

1. Are the proteins we use in fact present basally in normal tracheal cells, i.e. are they good candidates for transcytosed cargo?
2. Do they change their localisation when endocytosis is disrupted? And this point can be divided into two aspects: a. do they change at the basal membrane? b. do they change at the apical membrane (this latter point is not questioned the referee)?

1. The FGFR and beta-integrin are the only known basal markers in tracheal cells. A major reason for being confident of their presence in the basal membrane, even though they are difficult to visualize, is that the biological function of both is at the basal membrane, with the FGFR receiving growth or chemotactic signals from the surrounding tissue, and integrins anchoring the branches on the underlying tissue. However, it is indeed the case that their expression levels are very low, and it is difficult to visualize them, whether by expression of GFP-labelled constructs or by immunofluorescence. We have pushed to the limit a number of methods to improve the detection, but we seem to be constrained by the biology of these molecules.

In addition to the low but detectable signal at the outer boundary cell, some signal is always visible within the cell, which we had in the past always interpreted as an artefact or background, but for which our findings here might provide alternative interpretations.

2. a. We agree with the referee's assessment that FGFR::GFP is still detectable in the basal membrane after blocking endocytosis. This is, in our view, no contradiction to our model. The most parsimonious interpretation is that this is the FGFR that had already been delivered basally before we interfered with endocytosis, and which remains there after endocytosis is blocked.

b. In addition to this basal pool of FGFR, cells with blocked endocytosis accumulate abnormally high levels of FGFR at the apical membrane, in fact at much higher intensities than at the basal membrane. This is the more dramatic aspect of the phenotype, and our conclusions therefore rely not so much on a possible reduction of basal signal after blocking endocytosis (which would not be possible to demonstrate reliably), but rather on the abnormally enriched presence in the apical membrane.

A technical point: The increase in FGFR::GFP on endocytosis blockage that we show in Figure S4 corresponds to the cytoplasmic + apical pool of the FGFR. We used the Dof signal in the cell to create a mask of the total cell volume to 3D-segment the FGFR signal. Therefore, this analysis does not take into account the FGFR that is present in the basal membrane. We had explained this only in the methods section but will now describe it more explicitly in the Results section.

On the other hand, and very importantly, how does FGFR localization relates to its activity? The authors show that when endocytosis is prevented dpERK (i.e. a reporter of FGFR activation) is not decreased, indicating that FGFR is normally active. Wouldn't this suggest that FGFR is still localized basally to receive Bnl signal?

Indeed, and this is also what we see. This is not in conflict with any of the results or known functions of the receptor. If endocytosis is blocked, FGFR cannot be internalized and removed from the basal membrane, where it is needed to receive the FGF from the surrounding cells.

Our concern, and the reason for doing the experiment, had been that endocytosis might be *required* for FGF signaling, and that this might account for the failure of the cell to grow. But it turns out that our results show that this is not the case, at least not for the terminal cell in this time frame.

Actually, as the authors indicate, in larval tracheal cells the intracellular accumulation of the FGFR leads to a reduced FGF signal transduction (Chanut-Delalande et al., 2010), suggesting that reduced FGF signaling activity in these cells is due to less FGFR reaching the basal membrane.

That is true, and again, not inconsistent with our own results. In the cited study, trafficking was blocked at a step downstream of endocytosis. In this experimental situation, the internalisation of the FGFR would therefore occur as normal in these cells, but due to the impaired function of the ESCRT complex, intracellular processing, and therefore potential re-delivery to the basal membrane would be impaired. Furthermore, if (as we now propose) FGFR is also delivered via late endosomes in this context, blocking the ESCRT pathway should also impair initial FGFR delivery. In either way, initial delivery or re-delivery of the receptor being blocked, it is reasonable to assume that reduced signal transduction is the result of reduced basal FGFR.

Thus in our study we see no reduction in basal FGFR and no reduction (and even an increase) in signaling, while Chanut et al see reduced basal FGFR and reduced signaling, and the reason for this is that they interfere with a different step of the membrane trafficking pathway.

The results with Myospheroid are not very convincing either, as the authors just show a single confocal section of control and shibire mutants.

We observed this phenotype in several instances. We will quantify it for the resubmission.

In summary, I consider that this very important point needs to be better documented before concluding that apical membrane material containing basal cargoes is transcytosed to the basal membrane.

Another conclusion that, to my opinion, should be better explained and documented, is the coordination of tube and basal membrane growth. Following the movements of the vesicles the authors conclude that there is a net displacement of these vesicles to the tip of the cell. This correlates with the presence of mature endosomes there. So the results postulate that transcytosis occurs at the tip, and therefore the growth of the basal membrane would occur preferentially at the tip. It has been demonstrated, as the authors indicate, that the tube membrane grows all along the length of the tube. How is then coordinated the tube and basal membrane growth? If, as the authors propose in their model, the tube membrane also grows after a process of endocytosis and recycling, wouldn't it be expected to have preferential tip growth? How do the authors reconcile all these observations with previously published results (Gervais and Casanova, 2010)?

The elegant study by Gervais and Casanova (2010) used a very clever method, which was however entirely non-quantitative (and did not make any claims to the contrary, either). The conclusion that material is delivered to the tube throughout its length was based on looking at the displacement relative to the base and the tip of the cell of short and transient secondary branches (seen for example in Fig. 1H in our paper). If the tube would primarily receive material at the tip, these secondary branches would not change their position with respect to the base of the cell. Instead, these branches tend to be displaced towards the tip, which shows that material is also added between the branch and the base of the cell. These branches are seen only in a fraction of wild type terminal cells and quantification is therefore difficult. Thus, the experiment shows convincingly that material is also delivered behind the transient branch and excludes a model by which all growth occurs only at the tip. But it does not discuss what proportion of the total is delivered along the length vs the tip of the branch.

Our model also does not contest the idea of ubiquitous membrane delivery over the length of the tube, either in the initial delivery step, or during redistribution. On the contrary, the presence along the length of the cell of vesicles carrying FYVE::GFP and Rab7, and of smaller MVB-like bodies in the EM sections, suggest that the pathway can also be deployed at a distance from the tip.

The serial-section electron tomography analysis is very interesting and identifies different sorts of vesicles. However, it is very unclear what the different vesicles referred in the models correspond to (in the in vivo imaging for instance). For instance, the small granular or the dense-core vesicles correspond to endocytic vesicles at different stages of maturation?

Based on distribution alone, it is almost impossible to determine which of these vesicles correspond to endosomes or to secretory vesicles, even for the extremely experienced EM experts in the team. We would require high resolution CLEM, and a wide range of fluorescent markers to be able to determine which population of vesicles found on EM correspond to each marker. Due to the broad distribution of these vesicles within the cell and their small sizes right now it would be extremely technically challenging to pursue this question (even though we too would love to know)

If there is a constant endocytosis from the apical membrane to generate the basal and build the definitive apical membrane, wouldn't it be expected to find many more vesicles around the tube? Wouldn't it be expected to find coated vesicles around or budding from the tube, as the coated vesicles observed budding from the basal membrane in Fig 2D? Or is the endocytosis observed mediated by non-clathrin coated vesicles?.

We agree, and we had observed this to be the case, but had not quantified this. We have now analysed the distribution of coated pits and their density in the apical or the basal membrane of the cell. Overall, we found a higher density of endocytic events in the apical membrane than in the basal membrane. As the reviewer noticed, we also found that the majority of endocytic events in the basal membrane occur towards the tip of the cell. We will add these data to Figure S3.

- Should the authors qualify some of their claims as preliminary or speculative, or remove them altogether?

The results of Serp accumulation upon MVB interference can lead to confusion (from line 404). The authors seem to suggest that Serp protein is exclusively produced in the FB and transported by transcytosis to reach the tracheal lumen. However, Serp is also produced in the tracheal cells themselves. In fact, serp expression in the FB seems to be detected by late embryogenesis, while expression in tracheal cells is detected much earlier (Dong et al. 2014; Luschnig et al. 2006; Wang et al. 2006). It was also shown that Serp undergoes a recycling mechanism from the lumen to the lumen, through the endosomes-TGN retrograde trafficking, that may also require Shrub (Dong et al, 2014; Dong et al 2013). Thus, it is unclear (and even unlikely) that the Serp found in the vesicles in Shrub-GFP mutants is derived exclusively from the transcytosed component from the FB. I suggest to better explore this issue or to remove this part.

We agree that the notion that the Serp we see is exclusively derived from the fat body is not correct. We observed Serp accumulation around Shrb::GFP sites in embryos at early and late stages, so it is likely that what we see is the result both of apical-to-apical redelivery of Serp (as reported in Dong et al., 2014a), and transcytosis of Serp from the basal membrane to the apical. We will therefore rewrite this. But regardless of whether we are looking at transcytosis, or apical-to-apical cycling, this experiment still reinforces the idea of our work that late endosomes serve as stations that collect material from and re-deliver it towards various compartments in the cell.

- Would additional experiments be essential to support the claims of the paper? Request additional experiments only where necessary for the paper as it is, and do not ask authors to open new lines of experimentation.

As indicated before, more conclusive results for transcytosis should be provided.

I suggest that the authors determine the presence of apical and basal cargoes (FGFR) in the late endosomes found when Shrub activity is impaired. According to his model, both types of markers should accumulate there. A high accumulation of these markers in those endosomes would reinforce the hypothesis proposed.

We agree with the reviewer that this would be a great experiment. Since we only have an FGFR tagged with GFP we cannot do this experiment using the Shrb::GFP line, so instead we would have to use *shrb* mutants. This in itself is not a problem (see also below – we will be adding some data), but the experiment would require multi-generation crosses which could only be started once the current Covid-19 restrictions are lifted and labs opened again. Instead, we propose to cite the following supporting evidence. Chanut-Delalande et al., (2010) showed that mutants for components of the ESCRT pathway *hrs* and *stam* show intracellular accumulation of overexpressed FGFR::GFP in tracheal cells of the air sac primordium, and Dong et al., (2014a) show that *shrb* mutants accumulate Crb in late endocytic compartments in tracheal cells of the dorsal trunks. We would suggest it is very likely that terminal cells therefore also accumulate Crb and FGFR in late endosomes in the absence of Shrb. An experiment would be nicer, but we fear this is the best we can do at the moment.

While it has been reported that *shrub*-GFP act as a dominant negative in different contexts (Dong et al, 2014; Sweeney et al 2006) it is unclear why. So it would be desirable to confirm the results with a loss of function condition (either mutant or RNAi line)

We will now add data on *shrb* mutants where we find a phenotype that is similar as in Shrb::GFP overexpression.

- Are the data and the methods presented in such a way that they can be reproduced?

The materials and methods section would benefit from more detailed explanations.

- Are the experiments adequately replicated and statistical analysis adequate?

Many of the experiments presented in this work are technically very challenging, like the in vivo analyses and particularly the serial-section electron tomography. This prevents having high numbers of replicates on occasions.

****Minor comments:****

-In the abstract the authors state: "We show that apical endocytosis and late endosome-mediated trafficking determine the membrane allocation to the apical and basal membrane domains". I think that the authors show that "that apical endocytosis and late endosome-mediated trafficking is required for correct membrane growth", but I am not that sure that they show that it determines the membrane allocation

We agree and will change this.

-References for the PH-GFP localization in cells should be provided. Which is the evidence that it only localizes to plasma membrane?

Apical localization of PH-GFP is *preferential* rather than *exclusive*. The construct has nevertheless been widely used as an 'apical' marker, based on the fact that this PH domain binds to PIP2, which is enriched in the apical domain of epithelial cells (e.g. Pilot, et al., 2006; Román-Fernández, et al., 2018)

-It would be more adequate to always use the same terms to facilitate the reading. For instance, in several figures the membranes are referred as basal plasma membrane and tube membrane, but in others outer and inner membrane

We will go through the entire manuscript and use a consistent nomenclature.

-Figure 3C,D and corresponding text are difficult to understand. The increase of fluorescence of the inner membrane seems to be very high, even higher than the corresponding to the outer membrane. Can the authors explain better this point and also describe better the method applied in the materials and methods section?

Rather than absolute amounts the graphs show fold increase over the amount of membrane at the beginning of the recording. We had used this representation for two reasons. First, the overall signal intensity can vary from one imaging set to the next, so comparing and representing absolute amounts from different datasets is not easily possible. However, we understand now why the representation in our plots was confusing. We will now show in Figure 3D how the total amount of plasma membrane in *shibire^{ts}* cells increases in the same way as in controls. For Figure 3C, we have found a better way of representing what percentage of the total membrane is present in each compartment as the cell grows. We will rewrite this part to make it clearer, and we will describe more thoroughly in the methods section how the analysis was done.

Reviewer #1 (Significance (Required)):

This work represents an important advance for the field for several reasons. First of all it represents a technical advance because the authors are able to combine the traditional genetic analysis with two powerful techniques (in vivo imaging and serial-section electron tomography) to analyze single cell behavior at high resolution (temporal and spatial). In addition it represents a conceptual advance as it proposes a mechanism through which membrane growth is coordinated to regulate cell morphogenesis. The mechanism presented (endocytosis and transcytosis) is not new but they find evidence in an in vivo system.

It was previously known that tracheal terminal cells undergo a process of intracellular tube formation and cell elongation at the same time, but the mechanisms coordinating these two cell events were not known. The proposed mechanism may not only be relevant for the morphogenesis of tracheal terminal cells, but could represent a general mechanism of cell morphogenesis. Therefore, the paper should be relevant for research in

the morphogenesis area but also in the cell biology field, as it shows how regulated membrane trafficking can control tissue morphogenesis

REFEREES CROSS-COMMENTING:

I agree with reviewer #4 on her/his comments and suggestions about analyzing the involvement of the recycling endosome in the process.

Reviewer #2 (Evidence, reproducibility and clarity (Required)):

In their manuscript, Mathew et al present a model in which transcytosis is utilized to deliver endocytosed apical membranes to supply basal membrane growth. The authors examined the developing terminal cell of the fruitfly tracheal system, which is a well established tubulogenesis model, as these cells form subcellular tubes by apical plasma membrane invagination. The authors show that basal membrane growth stops when endocytosis or endosomal transport is blocked, while the apical membrane grows excessively or membrane material accumulates in the cytosol, respectively.

Reviewer #2 (Significance (Required)):

The authors used high-end microscopy (including CLEM and electron tomography) to support their model and in my opinion, the quality and the quantity of the presented data are indeed adequate for this. The text is well written, the figures are of superb quality, and several cartoons help to understand the presented data/experiments. Therefore I highly recommend submitting the manuscript to a cell biology journal in its present form.

Reviewer #3 (Evidence, reproducibility and clarity (Required)):

This paper investigates the role of membrane trafficking in growth of the polarized tracheal cell that forms a cellular projection containing a subcellular tube. The authors show that apical endocytosis and late endosome-mediated trafficking determine the membrane allocation to the apical and basal membrane domains. Basal plasma membrane growth stops if endocytosis is blocked, whereas the apical membrane grows excessively. Plasma membrane is initially delivered apically, and then appears to be continuously endocytosed, together with apical and basal cargo. The sorting and recycling of apical and basolateral membrane appears to occur in a novel organelle carrying markers of late endosomes and multivesicular bodies (MVBs). Inhibiting endocytosis eliminates this compartment.

The work in this paper is generally of high quality, and with one exception, quite comprehensive. The writing and figures are clear.

****Major concerns:****

-A central focus of the paper is that balance between apical and basal membrane and the role of transcytosis in moving membrane from the apical compartment to the basolateral compartment. The current view is that transcytosis in mammalian cells usually goes through the recycling endosomes which are marked by rab11, although there is evidence for some trafficking through MVBs as well. In *Drosophila*, Rab11 positive recycling endosomes are frequently examined as part of endocytic system analyses. However, rab11 is not used as a marker in this paper and indeed there is no mention at all of recycling endosomes, even though recycling is at the core of the work. Since the authors do not examine rab11 or other possible markers of recycling endosomes, it is unclear whether the organelle they identify as carrying markers of late endosome and MVBs is some MVB/recycling endosome hybrid, or whether the organelle is completely distinct from the recycling endosome. Consequently, it is not possible to assess whether the observed trafficking either uses or does not use involve the recycling endosome. This ambiguity make is difficult to relate the observed trafficking to other systems. Minimally the authors should stain for rab-11 in WT and in some of the conditions where trafficking is perturbed

and determine if the MVB-like compartment they are observing is rab11 positive, and whether the recycling endosome are affected by the perturbations. Further experiments may be needed to resolve whether any trafficking is going via the recycling endosome or this new MVB-type structure, but without even preliminary data on the relationship between the MVB compartment and the recycling endosome, its hard to say what might be appropriate or exactly how long addressing this will take. But just staining for rab11 in WT and a few mutant conditions to get a handle on what is up with the recycling endosomes in these cells should take less than a month.

We had done a number of experiments on Rab11 but did not include them because we felt they did not add any crucial insights on the mechanism we describe here. However, as the reviewer rightfully points out, Rab11 is a classical marker for recycling and transcytosis and we agree that the reader should know our results. We found that overexpressed Rab11::GFP as well as endogenously tagged Rab11:YFP are both highly enriched around the tube. Unlike Rab5 which is seen in widely spaced discrete vesicles, Rab11 forms a cloud of small puncta. We found very low overlap of Rab11 with CD4::mIFP-positive vesicles at the tip of the cell. This suggests that Rab11 is unlikely to be directly involved in the transcytosis pathway we describe here.

Loss of Rab11 was harder to analyse; Rab11-RNAi did not show any obvious phenotype, which could be due to high maternal contribution or to low knockdown efficiency, none of which we analysed in detail. Expression of a dominant negative Rab11 resulted in very early defects (reported by Le Droguen et al., 2015) which prevented us from analysing the role of Rab11 in tube formation. But because Rab11 does not localize to the compartment at the tip of the cell, we believe that this structure does not rely on Rab11 to transfer material from the apical to the basal membrane of the cell.

We will add the data on Rab11 distribution in Figure S6.

-In addition to the above, I would recommend more discussion of how the authors' results relate to membrane trafficking and transcytosis in other systems. The recycling endosome should be considered, and it may be appropriate to draw comparisons to membrane trafficking in neurons that goes through MVBs (e.g. reviewed in VON BARTHELD and ALTICK Prog Neurobiol. 2011 Mar; 93(3): 313-340.). Although neurons are not hollow, they have definite morphological resemblance to tracheal terminal branches.

We thank the reviewer for the observation and will expand the discussion on MVB-mediated transcytosis in other systems.

-line 204-208 "To test whether raised levels of Crb were responsible for the excessive apical membrane, as reported in other contexts (Pellikka et al., 2002; Schottenfeld-Roames et al., 2014; Wodarz et al., 1995), we knocked down Crb (Fig. S4G-H)." According to the legend for Fig s4, the authors express an RNAi construct against *crbs*. However, there does not appear to be any quantification of the amount knockdown of *crb* that was achieved. This is a concern for two reasons: 1) RNAi in the embryonic trachea works poorly for most genes (for unknown reasons) 2) This does not appear to be a clonal experiment but rather a pan tracheal driving of Crb RNAi. Loss of *crbs* would be expected to have very negative effects on tracheal morphogenesis (although this hasn't been rigorously tested to this reviewer's knowledge), but there doesn't appear to be any adverse effects of pan-tracheal *crbs* RNAi, suggesting that little if any knockdown of *crb* was actually achieved.

The authors either need to document the reduction of crbs or remove this paragraph. Preferably, they would be able to document the reduction of crb because they are trying to address an important point and if they can show the apical expansion is crb-independent, that would be a nice result.

Loss of Crumbs definitely is detrimental to embryonic epithelia, to different degrees (Tepass and Knust, 1990). We therefore could not use mutants but expressed an RNAi, which does not abolish Crb completely. We have now determined the degree of *crb* knockdown in our experiments. We stained Crb in embryos that expressed *crb-IR* in the entire tracheal system (but leaving the epidermal expression intact) and quantified the amount of Crb in the tracheal dorsal trunks, normalized to the signal in the epidermis. We found that in *crb-IR* embryos, Crb levels were reduced by around 50% compared to control siblings. We will add these results to Fig. S4 in the new version of the manuscript.

Arrowheads point to Crb in the tracheal dorsal trunks, and brackets ([]) show the epidermis as a reference. Control embryos are siblings without *btl-gal4*, *UAS-PH::GFP*.

Minor concerns:

-Lines 250. "By this interpretation, unscissioned membrane invaginations protruding from the subcellular tube would occasionally have touched the basal plasma membrane or its protrusions and fused with it, as transcytosing vesicles would have done in the normal situation."

I am not convinced by the argument that the bridging invaginations are fusing analogously to transcytosing vesicles because a protrusion/nascent vesicle coming from the apical surface should have rab3 and V-SNAREs that should dock the protrusion/nascent vesicle with an endosomal compartment, not the basolateral surface. A transcytotic vesicle would have the rab3 and V-SNAREs for the basolateral membrane. So it would seem that a fusion of the apical surface directly to the basolateral surface would have to be an ectopic event outside of the normal situation.

We agree that the fusion events could also be unrelated to the normal physiology of the animal. In other contexts (e.g. the embryonic epidermis, the synaptic bouton) blocking dynamin results in long membrane invaginations as

a result of failure in membrane scission. In terminal cells, the apical and basal plasma membranes are very close to each other, and we believe this increases the chance of membrane invaginations meeting and fusion to take place. In addition, the long membrane invaginations we see seem to have been stripped of their clathrin coat suggesting that at least some aspects of 'vesicle' maturation proceed even though scission had failed. We also find evidence of small vesicles that resemble the contents of MVBs being deposited within the aberrant membrane invaginations. This suggests that MVBs are able to fuse with these unscissioned tubes and sheets, again indicating that the appropriate molecular markers are present, and the machinery in charge of generating these vesicles is active at the invaginated pits directly. In either case, we will rephrase our interpretations of these data and present it as speculation in the discussion section.

-Significant figures. This is not a big deal, but the authors are over reporting their significant figures. E.g. "a 7.05-fold increase (± 2.98 SD)". With an SD that is 50% the value of the measurement, reporting to hundreds is definitely beyond the accuracy of measurement. Rounding to tenths would be more appropriate.

We agree with the reviewer. We will rephrase this section and use more appropriate metrics. As prompted by Reviewer #1, we modified the analysis that corresponds to this sentence, which also modified the way data is normalized also reducing the spread. This happened because in the new analysis we compare apical and basal signal for each timepoint, which allows better comparison between different cells.

Reviewer #3 (Significance (Required)):

As there have not been that many studies on the dynamics of membrane trafficking during morphogenesis, the results should be of broad interest to those studying the endocytic system and the role of membrane trafficking in morphogenesis. However, the paper would be greatly strengthened if the authors considered the recycling endosome in their analysis and write up. As a well-known compartment for trafficking cargo and membrane to both the apical and basolateral surface, it is hard to know how to interpret the observed trafficking without knowing the involvement, or lack thereof, of recycling endosomes in this system.

Dear Ieva,

Daniel and I have now looked at this more carefully.

In particular, we of course discussed what could be done about the experiments on basal and apical cargo in MVBs. This presents a huge problem to us because of the current situation, which in the case of this project is particularly complex, for the following reasons.

- Daniel left EMBL just before the shutdown. As we wrote, he had fortunately been able to do some experiments already, and we will include the data. He had also begun to prepare for the longer-term experiments. However, because EMBL, unlike other institutions, opted for a complete shutdown, all ongoing fly crosses had to be terminated. So these crosses will have to be started from scratch.
- You obviously do not need any explanations on what it will take to start building the necessary stocks - either from small vials that have only been flipped here as rarely as possible by the two people at EMBL who were allowed in the fly room at all, or from the vials that Daniel took to Mexico. We are looking at 2 weeks expanding before we can even start collecting virgins, and then 3 generations of crosses.
- The crosses could be done at EMBL when it re-opens next week (though let's see what actually happens). But the problem is there is no longer anyone in my lab here who has the right expertise to stain and image the cells that Daniel had become an expert on.
- I had of course contingency plans in place for work on that project to be done after Daniel's departure, and in fact had planned to have a person here (or rather in Cologne, but no difference) with the right expertise. He was meant to arrive in early April, contract all sorted, BUT he is coming from India, and the German embassy there is in lockdown and won't even accept applications for visas, never mind issue visas, and heaven knows when he will be allowed out, and whether he then has to go into quarantine. Definitely nothing in sight before the end of June.
- Daniel himself is not allowed to work in Mexico, the university is definitely not opening before the end of May, and maybe even later.
- Renjith, the other first author, is also in lockdown in India, and even if we could send him the stocks, and the stocks survive, and he could set up the crosses, the imaging facilities there are up to the standard needed to reliably visualise and quantify the FGFR and integrin stainings (especially keeping in mind the low expression levels of the endogenous proteins).

The only other solution I could imagine is that we would do the crosses here, optimistically have stocks ready end of June, and Daniel flies over to do the analyses

here. But for the moment, no visitors allowed at EMBL, immigration restrictions, travel bans, no flights, etc etc. And no idea when and how the situation will change.

So, whichever way we turn, obstacles and problems. I think we're looking at earliest possible time for new imaging being end of July, provided the stainings (done in a new environment, newly set up lab) work at the first attempt, followed by data analysis etc. So, optimistic earliest time for resubmission would be August. And you know how such plans pan out....

In summary, even with the offer of the extension, I just don't quite see that we can be certain that we could obtain these results within a reasonable time in the current situation, if at all.

We do of course have all the other new supporting data, and data in the literature support our interpretations*, which we would cite prominently.

* quote from our response:

Chanut-Delalande et al., (2010) showed that mutants for components of the ESCRT pathway hrs and stam show intracellular accumulation of overexpressed FGFR::GFP in tracheal cells of the air sac primordium, and Dong et al., (2014a) show that shrb mutants accumulate Crb in late endocytic compartments in tracheal cells of the dorsal trunks. We would suggest it is very likely that terminal cells therefore also accumulate Crb and FGFR in late endosomes in the absence of Shrb. An experiment would be nicer, but we fear this is the best we can do at the moment.

Sorry, should have included Daniel's calculations on this; includes further technical challenges I had missed:

> The way I thought is the best way to address it is:

>

> 1) Getting a line with shrb mutation and FGFR::GFP = 3 generations (so that FGFR::GFP is homozygous)

>

> 2) Crossing it with shrb mutant line with btl>IFP and do live imaging = +1 generation.

>

> BUT this is just to see FGFR::GFP, we would then have to do a staining for Crb. The CD4::mIFP does not survive the fixation so we would also have to stain Dof to see the cell.

>

> Alternative:

>

> 1) Recombining btl>PIPcherry with FGFR::GFP = 3 generations

> 2) Double balancing it with shrb allele = + 3 generations

> 3) Staining it for Crb.

>

>

Dear Maria and Daniel,

With these calculations I agree that in the current situation it is too much work with an unpredictable outcome to wait for an extended period of time for these results. I would suggest to briefly describe these issues in the final point-by-point response and to include appropriate literature references and discussion as you indicated in your RC revision plan.

Second response to referees (our responses in blue font)

Reviewer #1 (Evidence, reproducibility and clarity (Required)):

****Summary:****

In this work the authors investigate the dynamics of cell morphogenesis in a convenient in vivo system. They use the terminal cells of the embryonic tracheal system and comprehensively address how cell shape change (elongation in this case) takes place and how membrane is remodeled during the process. By combining different high-resolution techniques, i.e. in vivo imaging of terminal cells expressing different membrane markers and serial-section electron tomography, they describe the organelle organization (ER, Golgi, different types of vesicles) in the terminal cells during their elongation. They identify the presence of membrane structures/vesicles particularly abundant at the tip of the cell ahead of the growing tube. When they block endocytosis they find increased tube membrane and lack of basal membrane growth. In addition, in tomograms, they observe clear membrane defects, like invaginations that could even connect the tube and basal membranes. This correlates with the absence of vesicles at the tip observed in normal conditions.

The analysis of the nature of the vesicles observed indicated the accumulation of late endosomes and MVB, particularly at the tip of the growing terminal cells. Interfering with the formation of these MVB led to defects in the growth of both the tube and the basal membrane.

Altogether the authors propose a model in which the newly formed membrane (and transmembrane proteins) passes through the ER and Golgi and reaches the apical membrane. The incorporated membrane is then rapidly endocytosed and follows a maturation pathway through MVB, from where different cargoes and membrane would be sorted and recycled back to the apical (tube) membrane, or to the basal membrane through a transcytosis mechanism

****Major comments:****

- Are the key conclusions convincing?

Most of the conclusions presented are convincing and supported by the results observed.

However, to my understanding, one of the key conclusions of this work (that membrane is transcytosed from the apical to the basal domain) is not fully convincing. A critical result to support the author's conclusion of apical-basal transcytosis is to find clear evidence of basal accumulation of a transcytosed marker.

The authors show accumulation of FGFR-GFP and Myospheroid as evidence. However, I find the results presented not very convincing. The accumulation of FGFR-GFP at the basal membrane in the control is not very clear in the images and movie presented. In addition, in the shibire mutant, some basal accumulation of FGFR-GFP seems to be detected (particularly in the movie). In the figures the authors show an increase of FGFR-GFP intensity when endocytosis is blocked, but this is not explained in the text.

If we understand the referee correctly, there are two parts to her/his concern:

1. Are the proteins we use in fact present basally in normal tracheal cells, i.e. are they good candidates for transcytosed cargo?
2. Do they change their localisation when endocytosis is disrupted? And this point can be divided into two aspects: a. do they change at the basal membrane? b. do they change at the apical membrane (this latter point is not questioned the referee)?

1. The FGFR and beta-integrin are the only known basal markers in tracheal cells. A major reason for being confident of their presence in the basal membrane, even though they are difficult to visualize, is that the biological function of both is at the basal membrane, with the FGFR receiving growth or chemotactic signals from the surrounding tissue, and integrins anchoring the branches on the underlying tissue. However, it is indeed the case that their expression levels are very low, and it is difficult to visualize them, whether by expression of GFP-labelled constructs or by immunofluorescence. We have pushed to the limit a number of methods to improve the detection, but we seem to be constrained by the biology of these molecules.

In addition to the low but detectable signal at the outer boundary cell, some signal is always visible within the cell, which we had in the past always interpreted as an artefact or background, but for which our findings here might provide alternative interpretations.

2. a. We agree with the referee's assessment that FGFR::GFP is still detectable in the basal membrane after blocking endocytosis. This is, in our view, no contradiction to our model. The most parsimonious interpretation is that this is the FGFR that had already been delivered basally before we interfered with endocytosis, and which remains there after endocytosis is blocked.

b. In addition to this basal pool of FGFR, cells with blocked endocytosis accumulate abnormally high levels of FGFR at the apical membrane, in fact at much higher intensities than at the basal membrane. This is the more dramatic aspect of the phenotype, and our conclusions therefore rely not so much on a possible reduction of basal signal after blocking endocytosis (which would not be possible to demonstrate reliably), but rather on the abnormally enriched presence in the apical membrane.

A technical point: The increase in FGFR::GFP on endocytosis blockage that we show in Figure S4 (now Figure EV2) corresponds to the cytoplasmic + apical pool of the FGFR. We used the Dof signal in the cell to create a mask of the total cell volume to 3D-segment the FGFR signal. Therefore, this analysis does not take into account the FGFR that is present in the basal membrane. We had explained this in the methods section (line 716), but in the main text we only referred to this experiment together with the rescue of FGFR::GFP distribution, which might have contributed to the confusion. We now refer to this experiment more explicitly in the Results section (line 308), and to the rescue experiment only later on.

On the other hand, and very importantly, how does FGFR localization relate to its activity? The authors show that when endocytosis is prevented dpERK (i.e. a reporter of FGFR activation) is not decreased, indicating that FGFR is normally active. Wouldn't this suggest that FGFR is still localized basally to receive Bnl signal?

Indeed, and this is also what we see. This is not in conflict with any of the results or known functions of the receptor. If endocytosis is blocked, FGFR cannot be internalized and removed from the basal membrane, where it is needed to receive the FGF from the surrounding cells.

Our concern, and the reason for doing the experiment, had been that endocytosis might be *required* for FGF signaling, and that this might account for the failure of the cell to grow. But it turns out that our results show that this is not the case, at least not for the terminal cell in this time frame.

Actually, as the authors indicate, in larval tracheal cells the intracellular accumulation of the FGFR leads to a reduced FGF signal transduction (Chanut-Delalande et al., 2010), suggesting that reduced FGF signaling activity in these cells is due to less FGFR reaching the basal membrane.

That is true, and again, not inconsistent with our own results. In the cited study, trafficking was blocked at a step downstream of endocytosis. In this experimental situation, the internalisation of the FGFR would therefore occur as normal in these cells, but due to the impaired function of the ESCRT complex, intracellular processing, and therefore potential re-delivery to the basal membrane would be impaired. Furthermore, if (as we now propose) FGFR is also delivered via late endosomes in this context, blocking the ESCRT pathway should also impair initial FGFR delivery. In either way, initial delivery or re-delivery of the receptor being blocked, it is reasonable to assume that reduced signal transduction is the result of reduced basal FGFR.

Thus in our study we see no reduction in basal FGFR and no reduction (and even an increase) in signaling, while Chanut et al see reduced basal FGFR and reduced signaling, and the reason for this is that they interfere with a different step of the membrane trafficking pathway as we explain in line 562.

The results with Myospheroid are not very convincing either, as the authors just show a single confocal section of control and shibire mutants.

We observed this phenotype in several instances. We have now quantified it and also included data on 2 hours of dynamin inactivation (Figure 6; line 306). We also added orthogonal views of the representative images to better show the phenotype we refer to.

In summary, I consider that this very important point needs to be better documented before concluding that apical membrane material containing basal cargoes is transcytosed to the basal membrane.

Another conclusion that, to my opinion, should be better explained and documented, is the coordination of tube and basal membrane growth. Following the movements of the vesicles the authors conclude that there is a net displacement of these vesicles to the tip of the cell. This correlates with the presence of mature endosomes there. So the results postulate that transcytosis occurs at the tip, and therefore the growth of the basal membrane would occur preferentially at the tip. It has been demonstrated, as the authors indicate, that the tube membrane grows all along the length of the tube. How is then coordinated the tube and basal membrane growth? If, as the authors propose in their model, the tube membrane also grows after a process of endocytosis and recycling, wouldn't it be expected to have preferential tip growth? How do the authors reconcile all these observations with previously published results (Gervais and Casanova, 2010)?

The elegant study by Gervais and Casanova (2010) used a very clever method, which was however entirely non-quantitative (and did not make any claims to the contrary, either). The conclusion that material is delivered to the tube throughout its length was based on looking at the displacement relative to the base and the tip of the cell of short and transient secondary branches (seen for example in Fig. 1H in our paper). If the tube would primarily receive material at the tip, these secondary branches would not change their position with respect to the base of the cell. Instead, these branches tend to be displaced towards the tip, which shows that material is also added between the branch and the base of the cell. These branches are seen only in a fraction of wild type terminal cells and quantification is therefore difficult. Thus, the experiment shows convincingly that material is also delivered behind the transient branch and excludes a model by which all growth occurs only at the tip. But it does not discuss what proportion of the total is delivered along the length vs the tip of the branch.

Our model also does not contest the idea of ubiquitous membrane delivery over the length of the tube, either in the initial delivery step, or during redistribution. On the contrary, the presence along the length of the cell of vesicles carrying FYVE::GFP and Rab7, and of smaller MVB-like bodies in the EM sections, suggest that the pathway can also be deployed at a distance from the tip.

The serial-section electron tomography analysis is very interesting and identifies different sorts of vesicles. However, it is very unclear what the different vesicles referred in the models correspond to (in the in vivo imaging for instance). For instance, the small granular or the dense-core vesicles correspond to endocytic vesicles at different stages of maturation?

Based on distribution alone, it is almost impossible to determine which of these vesicles correspond to endosomes or to secretory vesicles, even for the extremely experienced EM experts in the team. We would require high resolution CLEM, and a wide range of fluorescent markers to be able to determine which population of vesicles found on EM correspond to each marker. Due to the broad distribution of these vesicles within the cell and their small sizes right now it would be extremely technically challenging to pursue this question (even though we too would love to know)

If there is a constant endocytosis from the apical membrane to generate the basal and build the definitive apical membrane, wouldn't it be expected to find many more vesicles around the tube? Wouldn't it be expected to find coated vesicles around or budding from the tube, as the coated vesicles observed budding from the basal membrane in Fig 2D? Or is the endocytosis observed mediated by non-clathrin coated vesicles?.

We agree, and we had observed this to be the case, but had not quantified this. We have now analysed the distribution of coated pits and their density in the apical or the basal membrane of the cell. Overall, we found a higher density of endocytic events in the apical membrane than in the basal membrane. As the reviewer noticed, we also found that the majority of endocytic events in the basal membrane occur towards the tip of the cell. We added these data to current Figure 3 of the paper, and the description in line 235.

- Should the authors qualify some of their claims as preliminary or speculative, or remove them altogether?

The results of Serp accumulation upon MVB interference can lead to confusion (from line 404). The authors seem to suggest that Serp protein is exclusively produced in the FB and transported by transcytosis to reach the tracheal lumen. However, Serp is also produced in the tracheal cells themselves. In fact, serp expression in the FB seems to be detected by late embryogenesis, while expression in tracheal cells is detected much earlier (Dong et al. 2014; Luschnig et al. 2006; Wang et al. 2006). It was also shown that Serp undergoes a recycling mechanism from the lumen to the lumen, through the endosomes-TGN retrograde trafficking, that may also require Shrub (Dong et al, 2014; Dong et al 2013). Thus, it is unclear (and even unlikely) that the Serp found in the vesicles in Shrub-GFP mutants is derived exclusively from the transcytosed component from the FB. I suggest to better explore this issue or to remove this part.

We agree that the notion that the Serp we see is exclusively derived from the fat body is not correct. We observed Serp accumulation around *Shrb::GFP* sites in embryos at early and late stages, so it is likely that what we see is the result both of apical-to-apical redelivery of Serp (as reported in Dong et al., 2014a), and transcytosis of Serp from the basal membrane to the apical. We have rewritten this section (lines 449-462). But regardless of whether we are looking at transcytosis, or apical-to-apical cycling, this experiment still reinforces the idea of our work that late endosomes serve as stations that collect material from and re-deliver it towards various compartments in the cell.

- Would additional experiments be essential to support the claims of the paper? Request additional experiments only where necessary for the paper as it is, and do not ask authors to open new lines of experimentation.

As indicated before, more conclusive results for transcytosis should be provided.

I suggest that the authors determine the presence of apical and basal cargoes (FGFR) in the late endosomes found when Shrub activity is impaired. According to his model, both types of markers should accumulate there. A high accumulation of these markers in those endosomes would reinforce the hypothesis proposed.

We agree with the reviewer that this would be a great experiment. Since we only have an FGFR tagged with GFP we cannot do this experiment using the *Shrb::GFP* line, so instead we would have to use *shrb* mutants. This in itself is not a problem (see also below – have added some data), but the experiment would require multi-generation crosses which could only be started once the current Covid-19 restrictions are lifted and labs opened again. The experiment would imply re-expanding lines that had to be kept at minimum care during the lockdown, and at least three generations of crosses plus stainings and imaging. The two first authors of the paper are no longer in the lab and also have work limitations in their respective labs which might remain even for a longer time compared to Europe. And even in that case, the experiment would require shipping of lines and reagents. Given these conditions, instead, we propose to cite the following supporting evidence (now in line 558). Chanut-Delalande et al., (2010) showed that mutants for components of the ESCRT pathway *hrs* and *stam* show intracellular accumulation of overexpressed FGFR::GFP in tracheal cells of the air sac primordium, and Dong et al., (2014a) show that *shrb* mutants accumulate Crb in late endocytic compartments in tracheal cells of the dorsal trunks. We would suggest it is very likely that terminal cells therefore also accumulate Crb and FGFR in late endosomes in the absence of Shrb. An experiment would be nicer, but we fear this is the best we can do at the moment.

While it has been reported that *shrub-GFP* act as a dominant negative in different contexts (Dong et al, 2014; Sweeney et al 2006) it is unclear why. So it would be desirable to confirm the results with a loss of function condition (either mutant or RNAi line)

We have now added data on *shrb* mutants where we find a phenotype that is similar as in *Shrb::GFP* overexpression (Figure 8 and line 419).

- Are the data and the methods presented in such a way that they can be reproduced?

The materials and methods section would benefit from more detailed explanations.

We have explained our methods in greater detail (line 694 – 715).

- Are the experiments adequately replicated and statistical analysis adequate?

Many of the experiments presented in this work are technically very challenging, like the in vivo analyses and particularly the serial-section electron tomography. This prevents having high numbers of replicates on occasions.

Minor comments:

-In the abstract the authors state: "We show that apical endocytosis and late endosome-mediated trafficking determine the membrane allocation to the apical and basal membrane domains". I think that the authors show that "that apical endocytosis and late endosome-mediated trafficking is required for correct membrane growth", but I am not that sure that they show that it determines the membrane allocation

We agree and have changed this (line 50).

-References for the PH-GFP localization in cells should be provided. Which is the evidence that it only localizes to plasma membrane?

Apical localization of PH-GFP is *preferential* rather than *exclusive*. The construct has nevertheless been widely used as an 'apical' marker, based on the fact that this PH domain binds to PIP2, which is enriched in the apical domain of epithelial cells (e.g. Pilot, et al., 2006; Román-Fernández, et al., 2018). Our orthogonal views (Figure 6) also show its preferential membrane localization.

-It would be more adequate to always use the same terms to facilitate the reading. For instance, in several figures the membranes are referred as basal plasma membrane and tube membrane, but in others outer and inner membrane

We have gone through the entire manuscript and use a consistent nomenclature. We now use 'basal' throughout the document, only explaining at the beginning of the Results section that the basal membrane is topologically the outer membrane. In the case of the apical membrane, we avoided the use of 'inner membrane' and instead used 'apical' or 'tube', depending on the context.

-Figure 3C,D and corresponding text are difficult to understand. The increase of fluorescence of the inner membrane seems to be very high, even higher than the corresponding to the outer membrane. Can the authors explain better this point and also describe better the method applied in the materials and methods section?

Rather than absolute amounts the graphs showed fold increase over the amount of membrane at the beginning of the recording. We had used this representation for two reasons. First, the overall signal intensity can vary from one imaging set to the next, so comparing and representing absolute amounts from different datasets is not easily possible. However, we understand now why the representation in our plots was confusing. We have found a better way of representing the data, by showing the proportion of fluorescence material at the apical or at the basal membrane compartment (referred to in line 200). We also replotted the data as box plots to reflect more

accurately the spread of the data (Figure 4). We have also explained the calculations more thoroughly in the methods section (line 698).

Reviewer #1 (Significance (Required)):

This work represents an important advance for the field for several reasons. First of all it represents a technical advance because the authors are able to combine the traditional genetic analysis with two powerful techniques (in vivo imaging and serial-section electron tomography) to analyze single cell behavior at high resolution (temporal and spatial). In addition it represents a conceptual advance as it proposes a mechanism through which membrane growth is coordinated to regulate cell morphogenesis. The mechanism presented (endocytosis and transcytosis) is not new but they find evidence in an in vivo system.

It was previously known that tracheal terminal cells undergo a process of intracellular tube formation and cell elongation at the same time, but the mechanisms coordinating these two cell events were not known. The proposed mechanism may not only be relevant for the morphogenesis of tracheal terminal cells, but could represent a general mechanism of cell morphogenesis. Therefore, the paper should be relevant for research in the morphogenesis area but also in the cell biology field, as it shows how regulated membrane trafficking can control tissue morphogenesis

REFEREES CROSS-COMMENTING:

I agree with reviewer #4 on her/his comments and suggestions about analyzing the involvement of the recycling endosome in the process.

Reviewer #2 (Evidence, reproducibility and clarity (Required)):

In their manuscript, Mathew et al present a model in which transcytosis is utilized to deliver endocytosed apical membranes to supply basal membrane growth. The authors examined the developing terminal cell of the fruitfly tracheal system, which is a well established tubulogenesis model, as these cells form subcellular tubes by apical plasma membrane invagination. The authors show that basal membrane growth stops when endocytosis or endosomal transport is blocked, while the apical membrane grows excessively or membrane material accumulates in the cytosol, respectively.

Reviewer #2 (Significance (Required)):

The authors used high-end microscopy (including CLEM and electron tomography) to support their model and in my opinion, the quality and the quantity of the presented data are indeed adequate for this. The text is well written, the figures are of superb quality, and several cartoons help to understand the presented data/experiments. Therefore I highly recommend submitting the manuscript to a cell biology journal in its present form.

Reviewer #3 (Evidence, reproducibility and clarity (Required)):

This paper investigates the role of membrane trafficking in growth of the polarized tracheal cell that forms a cellular projection containing a subcellular tube. The authors show that apical endocytosis and late endosome-mediated trafficking determine the membrane allocation to the apical and basal membrane domains. Basal plasma membrane growth stops if endocytosis is blocked, whereas the apical membrane grows excessively. Plasma membrane is initially delivered apically, and then appears to be continuously endocytosed, together with apical and basal cargo. The sorting and recycling of apical and basolateral membrane appears to occur in a novel organelle carrying markers of late endosomes and multivesicular bodies (MVBs). Inhibiting endocytosis eliminates this compartment.

The work in this paper is generally of high quality, and with one exception, quite comprehensive. The writing and figures are clear.

****Major concerns:****

-A central focus of the paper is that balance between apical and basal membrane and the role of transcytosis in moving membrane from the apical compartment to the basolateral compartment. The current view is that transcytosis in mammalian cells usually goes through the recycling endosomes which are marked by rab11, although there is evidence for some trafficking through MVBs as well. In *Drosophila*, Rab11 positive recycling endosomes are frequently examined as part of endocytic system analyses. However, rab11 is not used as a marker in this paper and indeed there is no mention at all of recycling endosomes, even though recycling is at the core of the work. Since the authors do not examine rab11 or other possible markers of recycling endosomes, it is unclear whether the organelle they identify as carrying markers of late endosome and MVBs is some MVB/recycling endosome hybrid, or whether the organelle is completely distinct from the recycling endosome. Consequently, it is not possible to assess whether the observed trafficking either uses or does not use involve the recycling endosome. This ambiguity makes it difficult to relate the observed trafficking to other systems. Minimally the authors should stain for rab-11 in WT and in some of the conditions where trafficking is perturbed and determine if the MVB-like compartment they are observing is rab11 positive, and whether the recycling endosome are affected by the perturbations. Further experiments may be needed to resolve whether any trafficking is going via the recycling endosome or this new MVB-type structure, but without even preliminary data on the relationship between the MVB compartment and the recycling endosome, it's hard to say what might be appropriate or exactly how long addressing this will take. But just staining for rab11 in WT and a few mutant conditions to get a handle on what is up with the recycling endosomes in these cells should take less than a month.

We had done a number of experiments on Rab11 but did not include them because we felt they did not add any crucial insights on the mechanism we describe here. However, as the reviewer rightfully points out, Rab11 is a classical marker for recycling and transcytosis and we agree that the reader should know our results. We found that overexpressed Rab11::GFP as well as endogenously tagged Rab11::YFP are both highly enriched around the tube. Unlike Rab5 which is seen in widely spaced discrete vesicles, Rab11 forms a cloud of small puncta. We found very low overlap of Rab11 with CD4::mIFP-positive vesicles at the tip of the cell. This suggests that Rab11 is unlikely to be directly involved in the transcytosis pathway we describe here.

Loss of Rab11 was harder to analyse; Rab11-RNAi did not show any obvious phenotype, which could be due to high maternal contribution or to low knockdown efficiency, none of which we analysed in detail. Expression of a dominant negative Rab11 resulted in very early defects (reported by Le Droguen et al., 2015) which prevented us from analysing the role of Rab11 in tube formation. But because Rab11 does not localize to the compartment at the tip of the cell, we believe that this structure does not rely on Rab11 to transfer material from the apical to the basal membrane of the cell.

We added the data on Rab11 distribution on Figure EV4, described in line 356.

-In addition to the above, I would recommend more discussion of how the authors' results relate to membrane trafficking and transcytosis in other systems. The recycling endosome should be considered, and it may be appropriate to draw comparisons to membrane trafficking in neurons that goes through MVBs (e.g. reviewed in VON BARTHELD and ALTICK *Prog Neurobiol.* 2011 Mar; 93(3): 313-340.). Although neurons are not hollow, they have definite morphological resemblance to tracheal terminal branches.

We thank the reviewer for the observation. The proposed comparison enriched the discussion of the relevance of MVBs in physiology and morphogenesis (line 585).

-line 204-208 "To test whether raised levels of Crb were responsible for the excessive apical membrane, as reported in other contexts (Pellikka et al., 2002; Schottenfeld-Roames et al., 2014; Wodarz et al., 1995), we knocked down Crb (Fig. S4G-H)." According to the legend for Fig s4, the authors express an RNAi construct

against crbs. However, there does not appear to be any quantification of the amount knockdown of crb that was achieved. This is a concern for two reasons: 1) RNAi in the embryonic trachea works poorly for most genes (for unknown reasons) 2) This does not appear to be a clonal experiment but rather a pan tracheal driving of Crb RNAi. Loss of crbs would be expected to have very negative effects on tracheal morphogenesis (although this hasn't been rigorously tested to this reviewer's knowledge), but there doesn't appear to be any adverse effects of pan-tracheal crbs RNAi, suggesting that little if any knockdown of crb was actually achieved.

The authors either need to document the reduction of crbs or remove this paragraph. Preferably, they would be able to document the reduction of crb because they are trying to address an important point and if they can show the apical expansion is crb-independent, that would be a nice result.

Loss of Crumbs definitely is detrimental to embryonic epithelia, to different degrees (Tepass and Knust, 1990). We therefore could not use mutants but expressed an RNAi, which does not abolish Crb completely. We have now determined the degree of *crb* knockdown in our experiments. We stained Crb in embryos that expressed *crb-IR* in the entire tracheal system (but leaving the epidermal expression intact) and quantified the amount of Crb in the tracheal dorsal trunks, normalized to the signal in the epidermis. We found that in *crb-IR* embryos, Crb levels were reduced by around 50% compared to control siblings. We added these results to Figure EV2 in the new version of the manuscript and described them in line 240.

****Minor concerns:****

-Lines 250. "By this interpretation, unscissioned membrane invaginations protruding from the subcellular tube would occasionally have touched the basal plasma membrane or its protrusions and fused with it, as transcytosing vesicles would have done in the normal situation."

I am not convinced by the argument that the bridging invaginations are fusing analogously to transcytosing vesicles because a protrusion/nascent vesicle coming from the apical surface should have rab3 and V-SNAREs that should dock the protrusion/nascent vesicle with an endosomal compartment, not the basolateral surface. A transcytotic vesicle would have the rab3 and V-SNAREs for the basolateral membrane. So it would seem that a fusion of the apical surface directly to the basolateral surface would have to be an ectopic event outside of the normal situation.

We agree that the fusion events could also be unrelated to the normal physiology of the animal. In other contexts (e.g. the embryonic epidermis, the synaptic bouton) blocking dynamin results in long membrane invaginations as a result of failure in membrane scission. In terminal cells, the apical and basal plasma membranes are very close to each other, and we believe this increases the chance of membrane invaginations meeting and fusion to take place. In addition, the long membrane invaginations we see seem to have been stripped of their clathrin coat suggesting that at least some aspects of 'vesicle' maturation proceed even though scission had failed. We also find evidence of small vesicles that resemble the contents of MVBs being deposited within the aberrant membrane invaginations. This suggests that MVBs are able to fuse with these unscissioned tubes and sheets, again indicating that the appropriate molecular markers are present, and the machinery in charge of generating these vesicles is active at the invaginated pits directly. In either case, now present our interpretation of these data as speculation in the discussion section (line 518).

-Significant figures. This is not a big deal, but the authors are over reporting their significant figures. E.g. "a 7.05-fold increase (± 2.98 SD)". With an SD that is 50% the value of the measurement, reporting to hundreds is definitely beyond the accuracy of measurement. Rounding to tenths would be more appropriate.

We agree with the reviewer. We will rephrase this section and use more appropriate metrics. As prompted by Reviewer #1, we modified the analysis that corresponds to this sentence, which also modified the way data is normalized also reducing the spread. This happened because in the new analysis we compare apical and basal signal for each timepoint, which allows better comparison between different cells (line 204).

Reviewer #3 (Significance (Required)):

As there have not been that many studies on the dynamics of membrane trafficking during morphogenesis, the results should be of broad interest to those studying the endocytic system and the role of membrane trafficking in morphogenesis. However, the paper would be greatly strengthened if the authors considered the recycling

endosome in their analysis and write up. As a well-known compartment for trafficking cargo and membrane to both the apical and basolateral surface, it is hard to know how to interpret the observed trafficking without knowing the involvement, or lack thereof, of recycling endosomes in this system.

Thank you for submitting a revised version of your manuscript. I have now received reports from all of the original referees, who find that their main concerns have been addressed and are now broadly in favour of publication of the manuscript. There now remain only a few editorial issues that have to be addressed before I can extend formal acceptance of the manuscript:

1. Please check the edits and comments from our data editor in the figure legends section that require your input (the file is attached).
2. We require a Data Availability Section at the end of Materials and Methods. As far as I can see, no data deposition in external databases is needed for this paper. If I am correct, then please state in this section: "This study includes no data deposited in external repositories".
3. Figure panels for Fig. 6D, E, G and Fig 7H are not called out in the manuscript text.
4. Please add a short table of contents at the beginning of the Appendix and update the nomenclature to "Appendix Figure S1" etc.
5. Please remove movie legends from the manuscript file and zip together with the movie file as a plain text file (<https://www.embopress.org/page/journal/14602075/authorguide#expandedview>).
6. Scale bars are missing for Fig. EV3B and EV4F.
7. Thank you for submitting source data for your manuscript. Please separate the data in one file per figure - currently some tabs are not labelled, and it is unclear which figure they correspond to.
8. I would like to propose some alterations in the synopsis text that you kindly provided with the goal to streamline it and adjust to the journal style. I have also written a short blurb that will accompany the title of your study on our online table of contents. Please take a look at the proposed and let me know if any corrections or adjustments are necessary.

Please let me know if you have any further questions regarding any of these points. You can use the link below to upload the revised files.

Thank you again for giving us the chance to consider your manuscript for The EMBO Journal. I am looking forward to receiving the final version.

Referee #1:

The authors have addressed my past concerns. This is an interesting study. However, I still consider that more conclusive results for transcytosis would be desirable.

Referee #2:

In my opinion, the technical quality of this work is superb, and the advancement to understanding membrane trafficking in tracheal terminal branches and probably also in cells with similar shape (eg. neurons) is significant.

Referee #3:

This is a rereview of a revised version of a previously submitted paper. I think that the authors have responded adequately to concerns raised by the reviewers and the article is ready for publication.

The authors performed the requested editorial changes.

Thank you for addressing the final editorial issues. I am now pleased to inform you that your manuscript has been accepted for publication.

Please note that it is The EMBO Journal policy for the transcript of the editorial process (containing referee reports and your response letter) to be published as an online supplement to each paper. If you do NOT want this, please inform the Editorial Office via email immediately. More information is available here: http://emboj.embopress.org/about#Transparent_Process

Your manuscript will be processed for publication in the journal by EMBO Press. Manuscripts in the PDF and electronic editions of The EMBO Journal will be copy edited, and you will be provided with page proofs prior to publication. Please note that supplementary information is not included in the proofs.

Should you be planning a Press Release on your article, please get in contact with embojournal@wiley.com as early as possible, in order to coordinate publication and release dates.

Thank you again for this contribution to The EMBO Journal and congratulations on a successful publication!

Corresponding Author Name: Maria Leptin
Journal Submitted to: The EMBO Journal
Manuscript Number: EMBOJ-2020-105332 [RC-2020-00182]